# BEYOND TEXT-TO-IMAGE: LIBERATING GENERATION WITH A UNIFIED DISCRETE DIFFUSION MODEL

**Qingyu Shi**[1*], **Jinbin Bai**[2,4*†], **Zhuoran Zhao**[2], **Wenhao Chai**[3], **Kaidong Yu**[5], **Jianzong Wu**[1], **Shuangyong Song**[5], **Yunhai Tong**[1], **Xiangtai Li**[1‡], **Xuelong Li**[5‡], **Shuicheng Yan**[2‡]

[*]Equal Contribution, [†]Project Lead, [‡]Corresponding Authors

[1]Peking University, [2]National University of Singapore, [3]Princeton University, [4]Collov Labs, [5]TeleAI

✉: jinbin.bai@u.nus.edu

Model: https://huggingface.co/MeissonFlow/Muddit

Code: https://github.com/M-E-AGI-Lab/Muddit

## ABSTRACT

Unified generation models aim to handle diverse tasks across modalities—such as text-to-image generation and image-to-text generation—within a single architecture and decoding paradigm. Autoregressive unified models suffer from slow inference due to sequential decoding, and non-autoregressive unified models suffer from weak generalization due to limited pretrained backbones. We introduce Muddit, a **u**nified **d**iscrete **di**ffusion **t**ransformer that enables fast and parallel generation across both text and image modalities. Unlike prior unified diffusion models trained from scratch, Muddit integrates strong **visual priors** from a pretrained text-to-image backbone with a lightweight text decoder, enabling flexible and high-quality multimodal generation under a unified architecture. Empirical results show that Muddit achieves competitive or superior performance compared to models in both quality and efficiency. This work also highlights the potential of purely discrete diffusion, when equipped with strong visual priors, as a scalable and effective backbone for unified generation.

## 1 INTRODUCTION

Unified generative models have recently emerged as a promising paradigm for multimodal data, encompassing both text and images. Most existing approaches adopt the autoregressive (AR) framework (Touvron et al., 2023; Zhou et al., 2024), where modalities are represented as discrete token sequences and generated sequentially in raster order. While this paradigm is well-suited for language, it introduces severe inefficiencies in image generation: producing an image requires step-by-step prediction of thousands of tokens, leading to substantial computational cost. Moreover, the imposed rasterized order is poorly aligned with the inherently spatial structure of images. These limitations hinder speed/quality trade-offs and restrict flexible conditional generation, such as inpainting, thereby constraining the practical applicability of unified models in interactive or real-time scenarios.

To mitigate these issues, recent works (Chen et al., 2025a; Pan et al., 2025; Chen et al., 2025b) have proposed hybrid approaches that couple AR with diffusion-based image modeling methods (Ho et al., 2020; Esser et al., 2024), as shown in Fig. 1 (a). However, such "glue" architectures fall short of true unification, as they introduce additional complexity (e.g., special tokens in the template) into the training and inference pipeline. The gap between text and image modeling methods further increases the complexity, leaving current unified models without a principled multimodal generative paradigm.

As shown in Fig. 1 (b), recent work like Dual-Diffusion (Li et al., 2024c) explores unifying multimodal generation under the diffusion model, but it ultimately relies on continuous diffusion for image (Esser et al., 2024) and discrete diffusion for text (Swerdlow et al., 2025b; Shi et al., 2024). This fundamental mismatch in generative principles undermines its claim of a true unification paradigm.

1

Figure 1: We propose Muddit, the first unified discrete diffusion model with a visual prior. Compared to language prior models like Show-o (Xie et al., 2024), Muddit demonstrates strong performance at image captioning and visual question answering. It also delivers clearer gains (7.0 vs 3.0) in image generation over the visual prior model D-DiT (Li et al., 2024c).

UniDisc (Swerdlow et al., 2025a) takes a more promising step by applying discrete diffusion over multimodal token spaces[1]. This allows parallel refinement of text and image tokens, improving inference efficiency and enabling more flexible conditioning. However, the overall quality of UniDisc's generation remains far from satisfactory. For example, it fails to match the fidelity of early diffusion models such as Stable Diffusion v1.5 (Rombach et al., 2022), and lacks support for vision-language reasoning tasks such as visual question answering (VQA). We attribute these shortcomings to the pretrained model's lack of prior knowledge. Without modular components carrying rich priors, these models face generalization and scalability bottlenecks.

Taken together, the two dark clouds: ineffective unified paradigm and the lack of strong prior knowledge, highlight the need for a new generation of unified models. In this work, we present **Muddit**, a **M**askGIT-style **u**nified **d**iscrete **di**ffusion **t**ransformer equipped with a lightweight text decoder. By combining the strengths of parallel discrete diffusion and semantically rich visual priors from a pretrained Meissonic text-to-image backbone (Bai et al., 2025b), Muddit enables scalable, efficient, and flexible sampling while significantly improving alignment and quality across modalities and various tasks such as high-resolution text-to-image generation, image captioning, and visual question answering, as shown in Fig. 1 (c).

We systematically detail the training objective of unified discrete diffusion models, the masking strategy, and the shared inference sampling strategy across three tasks. Finally, we conduct comprehensive evaluations with current popular unified models on several benchmarks, including GenEval (Ghosh et al., 2024), CIDEr (Lin et al., 2014), VQAv2 (Antol et al., 2015), GQA (Hudson & Manning, 2019), MME (Fu et al., 2023), and MMMU (Yue et al., 2024), demonstrating Muddit's superior performance and efficiency, validating that the unexplored purely discrete diffusion approach can rival much larger autoregressive-based unified models. While concurrent unified generation models (Yang et al., 2025) often build upon a language modeling prior—leveraging pretrained dLLMs as the backbone—we instead take a visual-first approach. Muddit is built upon an image generation prior, offering a new path toward unifying vision and language tasks within a discrete diffusion framework. We hope that this work inspires a new trend for unified generative modeling, grounded in discrete diffusion, beyond the boundaries of traditional text-to-image generation (Bai et al., 2025b).

## 2 METHOD

### 2.1 DISCRETE DIFFUSION WITH UNIFIED IMAGE AND TEXT PERSPECTIVE

In discrete diffusion, a sample $x \in \mathcal{X}$ is treated as a one-hot vector $\mathbf{x}$, where $\mathcal{X} = \{1, \ldots, N\}$. For language models, $N$ equals the vocabulary size. While for image models, $N$ is the number of discrete image token IDs obtained from a tokenizer or VQ codebook. At each diffusion step, we stochastically corrupt the tokens, gradually transforming the data distribution into a maximally entropic categorical prior; the generative model then learns to invert this corruption. Following

---

[1]MaskGIT, MaskAR, RandomAR, and Discrete Diffusion share significant conceptual and practical overlaps, often differing only in decoding order or architectural nuances. We elaborate on their connections in the next section. While Meissonic (Bai et al., 2025b) follows the naming convention of MaskGIT (Chang et al., 2022), we standardize terminology in this paper by referring to all such models under the umbrella of Discrete Diffusion.

recent works (Lou et al., 2023; Bai et al., 2025b) that cast token corruption as a continuous–time Markov chain (CTMC) over the finite alphabet $\mathcal{X}$, we let

$$\frac{d\,p_t}{dt} = Q_t\,p_t, \tag{1}$$

where $p_t \in \mathbb{R}^{N+1}$ is the distribution of $x_t$, and the time–dependent matrix $Q_t$ transports the data distribution $p_0 \approx p_{\text{data}}$ to the maximally entropic "noise" distribution $p_1 = p_{\text{stationary}}$. We adopt the absorbing-state (masked) diffusion variant that has proved particularly effective in text modelling: every symbol can jump to a dedicated mask token $\mathtt{m} = (\underbrace{0,\ldots,0}_{N}, 1)$ but never leaves it, i.e. $\mathtt{m}$ is an absorbing class.

**Forward posterior.** Marginalizing $\mathbf{x}$ gives

$$q(x_t \mid \mathbf{x}) = \text{Cat}\big(x_t \mid \alpha_t \mathbf{x} + (1 - \alpha_t)\mathtt{m}\big). \tag{2}$$

$\text{Cat}(\cdot)$ denotes a categorical distribution; it returns a one-hot token sampled from the probability vector inside the parentheses. $\alpha_t \in [0, 1]$ is the *survival probability*, *i.e.* the probability that an individual token has not yet been masked by time $t$. Thus $x_t$ equals the original clean token with probability $\alpha_t$ and equals the mask token $\mathtt{m}$ with probability $1 - \alpha_t$.

**Reverse process.** For any $0 < s < t < 1$, the CTMC induces an analytic posterior

$$q(x_s \mid x_t, \mathbf{x}) = \begin{cases} \text{Cat}(x_s \mid x_t), & x_t \neq \mathtt{m}, \\ \text{Cat}\Big(x_s \mid \dfrac{(1 - \alpha_s)\mathtt{m} + (\alpha_s - \alpha_t)\mathbf{x}}{1 - \alpha_t}\Big), & x_t = \mathtt{m}, \end{cases} \tag{3}$$

$x_t$ and $x_s$ are the corrupted tokens at times $t$ and $s$ ($s < t$). If $x_t$ is already a real vocabulary token ($x_t \neq \mathtt{m}$) it stays unchanged going backwards; otherwise, when $x_t = \mathtt{m}$, the distribution over $x_s$ is a convex combination of the mask and the clean token $\mathbf{x}$, weighted by their respective survival probabilities $\alpha_s$ and $\alpha_t$.

**Training Objective.** We employ a masked-token predictor $x_\theta(x_t, \alpha_t) \approx \mathbf{x}$, which leads to the continuous-time negative ELBO (Ou et al., 2024; Sahoo et al., 2024; Shi et al., 2024)

$$\mathcal{L}_{\text{NELBO}} = \mathbb{E}_{q(x_t|\mathbf{x})} \Big[ \int_0^1 \frac{\alpha_t'}{1 - \alpha_t} \log\big(x_\theta(x_t, \alpha_t) \cdot \mathbf{x}\big)\, dt \Big], \tag{4}$$

where $\alpha_t' = \frac{d\alpha_t}{dt}$ and $\mathbf{x}$ is the one-hot vector of ground truth. $x_\theta(x_t, \alpha_t) \in \mathbb{R}^{N+1}$ is the model's predicted categorical probability vector for the clean token given the corrupted input $(x_t, \alpha_t)$.

During generation, we start from an all-mask sequence ($t = 1$) and integrate the reverse CTMC towards $t = 0$, repeatedly replacing every masked position with the model's categorical prediction. Because the corruption schedule and objective are *identical* for any discrete alphabet $\mathcal{X}$, the same diffusion backbone unifies text and image generation. In the following section, we present Muddit, a unified framework that leverages discrete diffusion to model the generation tasks for both text and image jointly.

## 2.2 MUDDIT

### 2.2.1 UNIFIED ARCHITECTURE

As shown in Fig. 2, our architecture comprises a text encoder $\mathtt{E}_{\text{txt}}$, image encoder $\mathtt{E}_{\text{img}}$, transformer generator $\mathtt{G}$, sampler $\mathtt{S}$, text decoder $\mathtt{D}_{\text{txt}}$, and image decoder $\mathtt{D}_{\text{img}}$. The generator $\mathtt{G}$ is a single MM-DiT model, following the dual-/single-stream design of FLUX (Labs, 2024). Importantly, the generator $\mathtt{G}$ is initialized from Meissonic (Bai et al., 2025b), which has been extensively trained for high-resolution text-to-image generation. This initialization brings in a strong pretrained image prior, capturing rich spatial structures and semantic correlations across image and text tokens, which significantly enhances sample quality and accelerates convergence in the multimodal setting. Consequently, the same MM-DiT predicts the masked tokens for both modalities, which produces a shared generator for text and image synthesis.

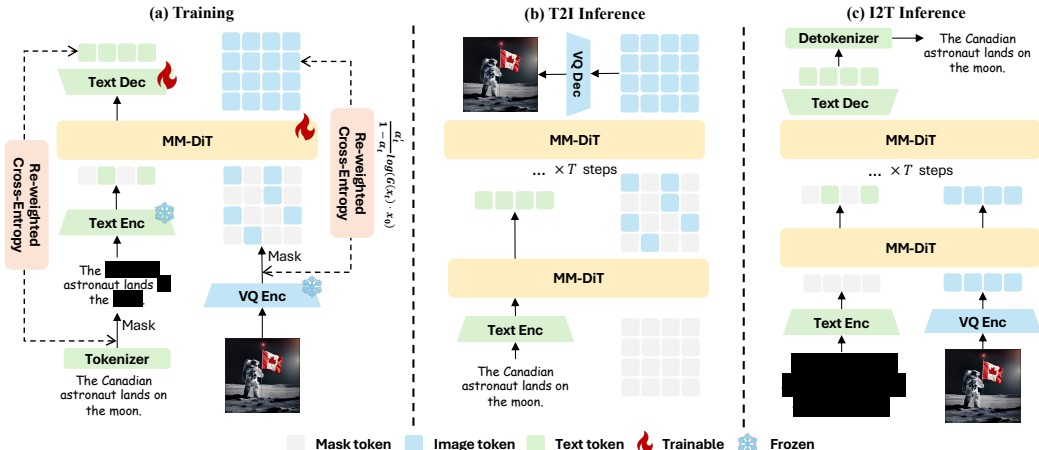

Figure 2: The training and inference architecture of Muddit. **(a)** During training, we randomly mask tokens from one of the two modalities. MM-DiT is trained to predict the masked tokens using a re-weighted cross-entropy loss, which jointly optimizes both the MM-DiT backbone and a lightweight text decoder. **(b)** In text-to-image inference, we initialize the image latent features using all-masked tokens and iteratively predict each latent token via MM-DiT. **(c)** In image-to-text inference, we similarly initialize all text tokens as masked and generate the text through the same iterative decoding process. Specifically for VQA tasks, we append mask token IDs to the end of the question and predict all masked token IDs as the final answer.

To reduce the computational cost of high-resolution imagery and lengthy captions, we quantize both modalities into a compact discrete space. A pre-trained VQ-VAE acts as the image encoder $E_{img}$, mapping pixels to codebook indices, while the CLIP text model, as $E_{txt}$, provides the text token embeddings. The MM-DiT predicts clean tokens in this shared space, which a lightweight linear head $D_{txt}$ converts back to text tokens.

### 2.2.2 UNIFIED TRAINING

**Masking strategy.** We model the forward posterior in Eq. 2 of both modalities using time-dependent hyperparameters $\alpha_t$, with the mask ratio defined as $\gamma_t = 1 - \alpha_t$. While BERT (Devlin, 2018) employs a fixed mask ratio of 15%, this setting is suitable for token completion but insufficient for generation. To support generative tasks, the design of $\gamma_t$ must satisfy the following criteria:

1. $\gamma_t$ must be a continuous function, bounded between 0 and 1, for $t \in [0, 1]$.
2. $\gamma_t$ should monotonically decrease with respect to $t$, with boundary conditions $\gamma_0 \rightarrow 0$ (initially clean data) and $\gamma_1 \rightarrow 1$ (masking all tokens).

Several strategies for masking and sampling have been proposed to meet these criteria (Chang et al., 2022). We adopt a *cosine scheduling strategy*. During training, a timestep $t \in [0, 1]$ is sampled from a truncated arccos distribution, with the density function:

$$\gamma_t = \frac{2}{\pi}(1 - (1 - t)^2)^{-\frac{1}{2}}. \tag{5}$$

During training, a mask ratio $\gamma_t \in [0, 1)$ is randomly sampled for each modality $\mathbf{x}_0$ (either image or text tokens), and the forward process (Eq. 2) is applied by randomly replacing clean tokens with mask tokens to obtain $x_t$.

**Unified training objective.** Let $\mathbf{c}$ denote the conditioning: the text embedding when synthesizing an image, or the image embedding when generating a caption. We randomly sample a mask ratio by Eq. 5. Then we corrupt the target sequence $\mathbf{x_0}$ (image or text tokens) with the CTMC described in Eq. 1 and train a single masked-token predictor $G(x_t, \alpha_t, \mathbf{c})$ to reconstruct $\mathbf{x_0}$. Both directions—text $\rightarrow$ image and image $\rightarrow$ text—share the identical continuous-time negative ELBO

$$\mathcal{L}_{\text{unified}} = \mathbb{E}_{q(x_t|\mathbf{x})}\left[\int_0^1 \frac{\alpha_t'}{1 - \alpha_t} \log\left(G(x_t, \alpha_t, \mathbf{c}) \cdot \mathbf{x}\right) dt\right], \tag{6}$$

where all symbols are as in Eq. 4 but the G now receives the cross-modal condition **c** as an additional input. **Key point:** switching from text → image to image → text merely changes the conditioning signal **c**; the loss Eq. 6 itself is unchanged. This symmetry keeps optimization identical across tasks and allows us to train a single parameter set jointly for both generation directions. During inference we again start from an all-mask sequence ($t=1$) and integrate the reverse CTMC towards $t=0$, feeding in the desired condition **c** to obtain either an image or a sentence from the same diffusion backbone.

### 2.2.3 UNIFIED INFERENCE

**Sampling strategy.** During inference, we apply the time-reversed posterior as defined in Eq. 3.

$$\mathtt{S}(\mathtt{G}, x_t, t) = p_\theta(x_s \mid x_t) = \begin{cases} \mathrm{Cat}(x_s \mid x_t), & x_t \neq \mathtt{m}, \\ \mathrm{Cat}\Big(x_s \mid \dfrac{(1-\alpha_s)\mathtt{m} + (\alpha_s - \alpha_t)\mathtt{G}(x_t, \alpha_t, \mathtt{c})}{1-\alpha_t}\Big), & x_t = \mathtt{m}, \end{cases} \tag{7}$$

where $\theta$ denotes the parameters of G, **c** is the multimodal condition, and $\alpha_t$ in Eq. 5 is applied sequentially with $t$ taking values $1, \frac{T-1}{T}, \ldots, \frac{1}{T}$, where $T$ is the total number of reverse steps. At each timestep $t$, Muddit predicts a fraction $\gamma_{t+\frac{1}{T}} - \gamma_t$ of the masked tokens by G and updates the masked tokens $x_t$ by S , continuing iteratively until all masked tokens are recovered. This dynamic approach offers several advantages over autoregressive methods, which require the model to learn conditional probabilities $P(x_i \mid x_{<i})$ based on a fixed token ordering. In contrast, random masking with a variable ratio enables the model to learn $P(x_i \mid x_\Lambda)$, where $\Lambda$ denotes an arbitrary subset of observed tokens. This flexibility is essential for parallel sampling, allowing multiple tokens to be predicted simultaneously rather than sequentially.

Our Muddit supports three tasks with a single generator G and sampler S: (i) text → image, (ii) image → text (captioning), and (iii) visual–question answering (VQA). The only change across tasks is the conditioning source **c** provided to G; the diffusion process and guidance logic are shared.

**(i) Text → image.** Given a text prompt $\mathtt{tp} \in \mathcal{T}$, the text encoder $\mathtt{E}_{\mathtt{txt}}$ produces a text token embedding $\mathtt{c}_{\mathtt{txt}} = \mathtt{E}_{\mathtt{txt}}(\mathtt{tp})$. Starting from a fully masked sequence $x_1$, the generator produces logits

$$l_t = \mathtt{G}(x_t, \alpha_t, \mathtt{c}_{\mathtt{txt}}), \qquad x_{t-\frac{1}{T}} = \mathtt{S}(l_t, x_t, t), \tag{8}$$

for $k = 1, \frac{T-1}{T}, \ldots \frac{1}{T}$. After $T$ steps we obtain visual tokens $x_0$, which the image decoder $\mathtt{D}_{\mathtt{img}}$ converts to a pixel-space image $I = \mathtt{D}_{\mathtt{img}}(x_0)$.

**(ii) Image → text.** For captioning, an input image $I \in \mathcal{I}$ is tokenized by the image encoder $\mathtt{E}_{\mathtt{img}}$: $\mathtt{c}_{\mathtt{img}} = \mathtt{E}_{\mathtt{img}}(I)$. The generator now conditions on the *visual* tokens while progressively decoding text:

$$l_t = \mathtt{G}(x_t, \alpha_t, \mathtt{c}_{\mathtt{img}}), \qquad x_{t-\frac{1}{T}} = \mathtt{S}(l_t, x_t, t), \tag{9}$$

yielding a text token sequence $x_0$, which $\mathtt{D}_{\mathtt{txt}}$ maps to a $\mathtt{caption} = \mathtt{Detokenize}(\mathtt{D}_{\mathtt{txt}}(x_0))$.

**(iii) Image + question → answer (VQA).** For visual–question answering we supply *both* an image and a question: $\mathtt{c}_{\mathtt{img}} = \mathtt{E}_{\mathtt{img}}(I)$ and $\mathtt{c}_{\mathtt{txt}} = \mathtt{E}_{\mathtt{txt}}(q)$. They are concatenated and fed to the generator, which outputs logits over answer tokens $x_k$:

$$l_t = \mathtt{G}(x_t, \alpha_t, [\mathtt{c}_{\mathtt{img}}, \mathtt{c}_{\mathtt{txt}}]), \qquad x_{t-\frac{1}{T}} = \mathtt{S}(l_t, x_t, t), \tag{10}$$

until the full answer $a$ is produced and decoded by $a = \mathtt{Detokenize}(\mathtt{D}_{\mathtt{txt}}(x_0))$.

**Classifier-free guidance.** At each decoding step, we apply the same guidance rule, independent of modality:

$$l_k \leftarrow \mathtt{G}(z_k, \alpha_k, \mathbf{c}) + \lambda\big[\mathtt{G}(z_k, \alpha_k, \mathbf{c}) - \mathtt{G}(z_k, \alpha_k, \mathbf{c}_{\mathrm{neg}})\big], \tag{11}$$

where $z_k$ (image or text tokens) is the partial target sequence, **c** is the *positive* condition (prompt, image, or image +question), $\mathbf{c}_{\mathrm{neg}}$ is the corresponding negative condition, and $\lambda$ is the guidance scale. Because the loss, decoding schedule, and guidance operator are *identical* in all three scenarios—only the conditioning signal changes—our framework realises a genuinely unified multimodal generator.

## 3 EXPERIMENT

### 3.1 EXPERIMENTAL SETUP

**Implementation details.** We build Muddit on top of the open-source Meissonic models (Bai et al., 2025b). The MM-DiT backbone is initialized with pretrained weights, and a lightweight linear head is added as a text decoder. Following Meissonic, we adopt the CLIP (Radford et al., 2021) as text encoder and VQ-VAE as image encoder and decoder, keeping them entirely frozen throughout all experiments. To support discrete denoising, we append a special <mask> token to CLIP's vocabulary for text masking, while the image mask token is inherited directly from Meissonic's initialization. We observe that, even without training, the <mask> embedding can already be predicted into a coherent sentence during training. Therefore, for simplicity, we freeze the <mask> embedding. During training, we use a constant learning rate of $1 \times 10^{-4}$ and a weight decay of $1 \times 10^{-2}$. Gradient accumulation is applied in both pretraining and supervised fine-tuning, resulting in an effective batch size of 1024. We train on 16 H100 GPUs for 5 days. During inference, we adopt the default Meissonic configuration, using cosine masking scheduling, 64 sampling steps, and a classifier-free guidance (CFG) scale of 9.0 and 1.5 for text-to-image and image-to-text generation, respectively.

**Training data.** We train Muddit in two stages using a combination of publicly available and internal datasets, including JourneyDB (Pan et al., 2023), LAION-Art (Schuhmann et al., 2022), CC12M (Changpinyo et al., 2021), and others. The final dataset is filtered based on aesthetic score, resolution, and aspect ratio, resulting in approximately 10 million image–text pairs. Both stages are optimized with the unified training objective defined in Eq. 6. Below, we describe the datasets and settings for each stage in detail.

1. **Pretraining.** We pretrain Muddit for 100K steps with a batch size of 1024, using the unified objective across both modalities. Text inputs are truncated to a maximum of 77 tokens, and images are resized to $512 \times 512$. The pretraining corpus consists of 8 million image–text pairs, recaptioned using Qwen2.5-VL-3B for improved consistency. Each batch is evenly split between text-to-image and image-to-text samples to enable joint training in both directions.

2. **Instruction tuning.** After pretraining, we fine-tune the model on a combination of 1 million instruction-following samples, including LLaVA-Instruct-150K, ALLaVA, SA-1B, and the VQAv2 training set. During this stage, only the answer portion of each prompt is masked. Additionally, we construct a curated dataset of 1 million high-quality image–text pairs to support multi-task training on VQA and image generation. Following the task instructions embedded in each sample, Muddit learns to produce long-form answers, concise replies, and image captions via task-specific prompting.

We present both quantitative and qualitative results for the T2I and I2T tasks in the following sections. Additional experiments and ablation studies are provided in the Appendix.

### 3.2 TEXT-TO-IMAGE GENERATION

**Quantitative results.** Following prior work, we evaluate our $512 \times 512$ model on GenEval (Ghosh et al., 2024) after supervised fine-tuning in Tab. 1. Muddit attains an overall accuracy of 0.61, surpassing prior discrete diffusion models such as Monetico (0.44) and Meissonic (0.54), and closely matching Stable Diffusion 3 (0.62) with only 1B parameters. It further shows strong compositional reasoning (0.72 on "Two Objects", 0.54 on "Counting"), and benefits from joint multimodal training, which enhances T2I performance. These results demonstrate the effectiveness of Muddit as the first unified discrete diffusion model for both text and image modalities.

**Qualitative results.** We present diverse generations from our model conditioned on rich textual prompts in Fig. 3. The outputs exhibit strong text-image alignment, capturing fine details in both realistic and imaginative scenes. Our model effectively renders complex structures, lighting, and textures across various domains.

### 3.3 IMAGE-TO-TEXT GENERATION

We present a comprehensive comparison of our model Muddit against other multimodal models across four benchmarks: MS-COCO (image captioning) (Lin et al., 2014), VQAv2 (Antol et al.,

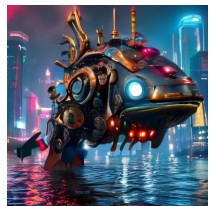

**Human**: A medieval town nestled on an expansive bay, cloaked in a blanket of snow and ice. Icy winds whip through cobblestone streets lined with half-timbered buildings. their stone walls adorned with intricate carvings. Steam rises from cozy hearths as villagers huddle around fires, their faces illuminated by lanterns. The sky is a deep indigo, flecked with stars as a full moon casts an ethereal glow over the scene.

**Human**: A towering, robotic whale sculpted with gears and cogs stands amidst a bustling futuristic cityscape bathed in neon light. Its colossal form is detailed with rusted brass accents and gleaming chrome fins that ripple as it glides across the shimmering water. Its eye stalks glow with an array of glowing LEDs.

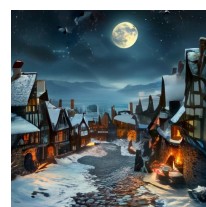

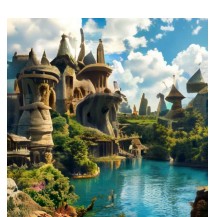

**Human**: An organic, bioluminescent pavilion, crafted from gleaming chrome and polished metal, rises from the tranquil depths of a misty ocean. Sunbeams pierce the swirling fog, illuminating shimmering structures in intricate curves and forms. The water below reflects an ethereal sheen of ambient light, creating a hazy, captivating scene.

**Human**: A high-quality photograph capturing the majestic Plum Bayou Mound Architecture in a vibrant city of anthropomorphic mammals. The architecture is intricate and detailed, with towering structures made of natural materials like stone and wood, set against a backdrop of lush greenery and vibrant blue water reflecting the afternoon sun.

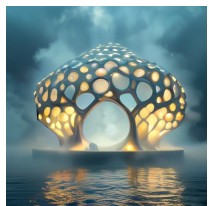

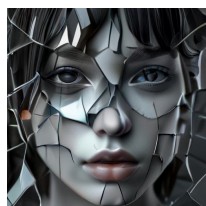

**Human**: A slice of creamy cheesecake, adorned with a vibrant yellow layer of lemon jelly and flecks of zesty lemon zest. The surface is smooth and inviting, captured in professional food photography with ultra-realistic detail. The lighting is soft, almost cinematic, highlighting the intricate texture and high definition sharpness.

**Human**: A 3D digitally rendered portrait in the style of "glass fragments" art. The face is depicted with a fractured, cracked surface reminiscent of shattered glass, using muted black and white tones. A split-toned effect adds depth and drama, highlighting the emotional intensity of the subject's expression. The image is ultra-high definition (4K) and rendered in Photoshop.

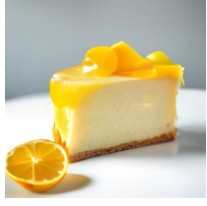

Figure 3: Samples of Text-to-Image Generation by Muddit.

**Human**: How many people are there in this image?

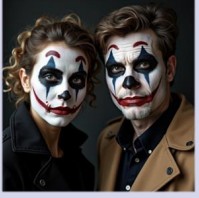

**Assistant**: There are two people in this image.

**Human**: What is the color of this mug?

**Assistant**: The color of the cup is white with blue picture on it.

**Human**: What is this statue made of?

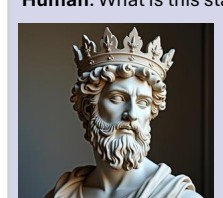

**Assistant**: The statue is made of marble.

**Human:** What is on the table?

**Assistant**: There are a cup, saucer, and a vase of flowers on the white table.

Figure 4: Samples of Visual Question Answering by Muddit.

2015), MME (Fu et al., 2023), GQA (Hudson & Manning, 2019), and MMMU (Yue et al., 2024) in Tab. 2. Notably, Muddit is the first unified model to employ discrete diffusion for both text-to-image and image-to-text generation, demonstrating that this approach is highly competitive.

**Quantitative comparison.** Muddit demonstrates strong performance across benchmarks despite its compact size (1B parameters). It achieves a CIDEr score of 59.9 on MS-COCO, surpassing diffusion-based baselines such as D-DiT (56.2). On VQAv2, it attains 68.2% accuracy, outperforming Show-O and D-DiT. Moreover, it reaches 1107.4 on MME and 57.5 on GQA, underscoring its competitiveness across multimodal reasoning tasks. These results highlight the effectiveness of Muddit as a unified diffusion-based model that balances efficiency with high-quality task performance.

Table 1: Evaluation of text-to-image generation performance on the GenEval (Ghosh et al., 2024).

| Model | Text Gen Arch | Image Gen Arch | Params (B) | Overall ↑ | Objects ↑ Single | Two | Counting ↑ | Colors ↑ | Position ↑ | Color ↑ Attribution |
|---|---|---|---|---|---|---|---|---|---|---|
| PixArt-$\alpha$ (Chen et al., 2024b) | - | Diffusion | 0.6 | 0.48 | 0.98 | 0.50 | 0.44 | 0.80 | 0.08 | 0.07 |
| SD 2.1 (Rombach et al., 2022) | - | Diffusion | 0.9 | 0.50 | 0.98 | 0.51 | 0.44 | 0.85 | 0.07 | 0.17 |
| DALL-E 2 (Ramesh et al., 2022) | - | Diffusion | 6.5 | 0.52 | 0.94 | 0.66 | 0.49 | 0.77 | 0.10 | 0.19 |
| SDXL (Podell et al., 2023) | - | Diffusion | 2.6 | 0.55 | 0.98 | 0.74 | 0.39 | 0.85 | 0.15 | 0.23 |
| DALL-E 3 (Betker et al., 2023) | - | Diffusion | - | 0.67 | 0.96 | 0.87 | 0.47 | 0.83 | 0.43 | 0.45 |
| SD 3 (Esser et al., 2024) | - | Diffusion | 2 | 0.62 | 0.98 | 0.74 | 0.63 | 0.67 | 0.34 | 0.36 |
| LWM (Liu et al., 2024c) | AR | AR | 7 | 0.47 | 0.93 | 0.41 | 0.46 | 0.79 | 0.09 | 0.15 |
| SEED-X (Ge et al., 2024) | AR | AR | 17 | 0.49 | 0.97 | 0.58 | 0.26 | 0.80 | 0.19 | 0.14 |
| Chameleon (Team, 2024) | AR | AR | 7 | 0.39 | - | - | - | - | - | - |
| Show-O (Xie et al., 2024) | AR | Discrete Diff. | 1.3 | 0.68 | 0.98 | 0.80 | 0.66 | 0.84 | 0.31 | 0.50 |
| Transfusion (Zhou et al., 2024) | AR | Diffusion | 8 | 0.67 | - | - | - | - | - | - |
| D-DiT (Li et al., 2024b) | Discrete Diff. | Diffusion | 2 | 0.65 | 0.97 | 0.80 | 0.54 | 0.76 | 0.32 | 0.50 |
| Monetico ($512 \times 512$) (Bai et al., 2025b) | - | Discrete Diff. | 1 | 0.44 | 0.92 | 0.48 | 0.26 | 0.78 | 0.06 | 0.13 |
| Meissonic ($1024 \times 1024$) (Bai et al., 2025b) | - | Discrete Diff. | 1 | 0.54 | 0.99 | 0.66 | 0.42 | 0.86 | 0.10 | 0.22 |
| UniDisc ($512 \times 512$) (Swerdlow et al., 2025b) | Discrete Diff. | Discrete Diff. | 1.4 | 0.42 | 0.92 | 0.47 | 0.15 | 0.67 | 0.13 | 0.19 |
| Muddit ($512 \times 512$) | Discrete Diff. | Discrete Diff. | 1 | 0.61 | 0.98 | 0.72 | 0.54 | 0.82 | 0.19 | 0.41 |

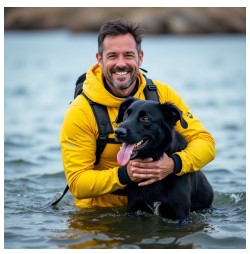

**Assistant**: The image depicts a man standing in water, holding a black dog and smiling. The man is wearing a yellow and wet jacket, and he has a happy expression. The background consists of a calm body of water with possibly a lake.

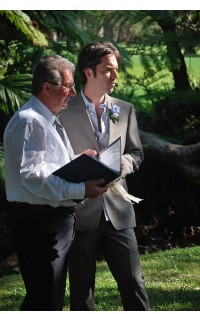

**Assistant**: The image depicts two individuals standing in a lush, forested area. the person on the left is older and wearing a light blue shirt and dark pants, while the person on the right is dressed in a suit with a shirt and tie. both appear to be engaged in a discussion or discussing something on the documents, with the person on the left holding a tablet.

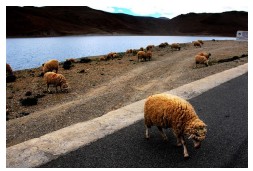

**Assistant**: The image depicts a group of sheep walking along a road near a body of water, possibly a lake or a body of water water, possibly a sea or sea. the road is lined with small rocks and a scattered vegetation, and a road is visible on the side of the road. in the background, there are rolling hills and a calm water with a fence.

**Assistant**: The image depicts a row of hot motorcycles parked in a parking lot. the motorcycle is a white model with a black and white color scheme, identifiable by its distinctive design and branding on the side. the motorcycle appears to be a modern model model, as evidenized by the branding and branding visible on the side. the background includes a residential building, some trees.

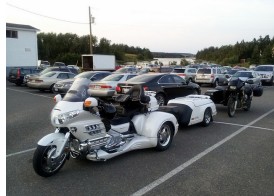

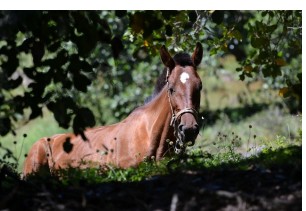

**Assistant**: The image depicts a brown horse standing amidst a lush, green forest. the horse appears to be a thoroughbred horse, characterized by its dense features and tail visible. it is equipped with a bridle and a decorative tag on its forehead. the background is filled with dense foliage and small plants, suggesting a natural, foreforested environment. the overall scene is serene.

Figure 5: Samples of Image-to-Text Generation by Muddit.

**Qualitative results.** We present example captions generated by our model across diverse scenarios in Fig. 5, including humans, animals, vehicles, and natural landscapes. The model demonstrates strong visual grounding and fine-grained descriptive ability, accurately capturing attributes such as clothing, expressions, background context, and object relationships. Fig. 4 illustrates our model's ability to accurately answer visual questions across various domains, including object counting, color recognition, material identification, and compositional reasoning.

## 3.4 ABLATION STUDY AND ANALYSIS

**Analysis of the inference timesteps.** As shown in Tab. 5, performance generally improves with more diffusion steps, plateauing around $T = 32$. GenEval and CIDEr see large gains from $T = 8$ to $T = 32$, with diminishing returns afterward. VQAv2 remains stable across timesteps, indicating

Table 2: Evaluation of image captioning, visual question answering on multimodal benchmarks.

| Model | Params (B) | Text Gen Arch | Image Gen Arch | MS-COCO CIDEr ↑ | VQAv2 Acc. ↑ | MME Acc. ↑ | GQA Acc. ↑ | MMMU Acc. ↑ |
|---|---|---|---|---|---|---|---|---|
| InternVL-2.0 (Chen et al., 2025e) | 8 | AR | - | - | - | 1648.1 | 61.0 | 49.3 |
| LLaVA-Next (Liu et al., 2024d) | 13 | AR | - | - | 82.8 | 1575.0 | 65.4 | 36.2 |
| BLIP-2 (Li et al., 2023) | 13 | AR | - | - | 65.0 | 1293.8 | 41.0 | 34.4 |
| QWEN-VL (Bai et al., 2023) | 7 | AR | - | - | 78.2 | 1487.5 | 57.5 | 35.9 |
| OpenFlamingo (Awadalla et al., 2023) | 9 | AR | - | 65.5 | 43.5 | - | - | 28.7 |
| Flamingo (Alayrac et al., 2022) | 9 | AR | - | 79.4 | 51.8 | - | - | - |
| Chameleon (Team, 2024) | 7 | AR | AR | 18.0 | - | - | - | - |
| LWM (Liu et al., 2024c) | 7 | AR | AR | - | 55.8 | - | - | - |
| Show-O (256×256) (Xie et al., 2024) | 1.3 | AR | Discrete Diff. | - | 64.7 | 1014.9 | 54.2 | - |
| Show-O (512×512) (Xie et al., 2024) | 1.3 | AR | Discrete Diff. | - | 69.4 | 1097.2 | 58.0 | 27.4 |
| Transfusion (Zhou et al., 2024) | 7 | AR | Diffusion | 29.0 | - | - | - | - |
| D-DiT (256×256) (Li et al., 2024b) | 2 | Discrete Diff. | Diffusion | - | 59.5 | 897.5 | 55.1 | - |
| D-DiT (512×512) (Li et al., 2024b) | 2 | Discrete Diff. | Diffusion | 56.2 | 60.1 | 1124.7 | 59.2 | - |
| UniDisc (Swerdlow et al., 2025b) | 1.4 | Discrete Diff. | Discrete Diff. | 46.8 | - | - | - | - |
| Muddit (512×512) | 1 | Discrete Diff. | Discrete Diff. | 59.9 | 68.2 | 1107.4 | 57.5 | 27.6 |
| Muddit (1024×1024) | 1 | Discrete Diff. | Discrete Diff. | 60.1 | 70.2 | 1139.2 | 57.8 | 28.7 |

Table 3: Impact of text loss weight. We apply the same text loss weight during both pretraining and instruction tuning.

| Benchmark | 0.2 | 0.4 | 0.6 | 0.8 | 1.0 |
|---|---|---|---|---|---|
| GenEval | 60.1 | 60.5 | **61.6** | 60.8 | 58.3 |
| MS-COCO | 51.4 | 52.1 | **59.9** | 58.8 | 59.4 |
| VQAv2 | 62.7 | 66.2 | 68.2 | 68.4 | **69.2** |

Table 4: Effect of joint training. We denote text-to-image as T2I and image-to-text as I2T, respectively.

| Benchmark | T2I only | I2T only | Joint training |
|---|---|---|---|
| GenEval | 59.3 | 28.3 | **61.6** |
| MS-COCO | - | **60.1** | 59.9 |
| VQAv2 | - | **69.1** | 68.2 |

that fewer steps suffice for discriminative tasks. Overall, a moderate number of steps provides a good balance between accuracy and efficiency.

**Analysis of the text loss weight.** As shown in Tab. 3, moderate text loss weights (around 0.6) yield the best overall performance. CIDEr and GenEval peak near this value, suggesting that both insufficient and excessive text weighting can harm generation quality. VQAv2 continues to improve with stronger text supervision but begins to plateau beyond 0.6. Overall, while discriminative tasks benefit from heavier textual guidance, generative tasks require a balanced mix of visual and textual signals—highlighting the importance of grounding language in multimodal learning.

Table 5: Performance across different diffusion timesteps.

| Sample steps | GenEval | CIDEr | VQAv2 |
|---|---|---|---|
| T=8 | 51.6 | 43.6 | 53.9 |
| T=16 | 58.5 | 59.3 | 57.4 |
| T=24 | 59.3 | 59.4 | 62.3 |
| T=32 | **61.9** | 59.7 | 65.4 |
| T=40 | 61.7 | 60.1 | 66.8 |
| T=64 | 61.1 | 59.9 | **68.2** |

**Analysis of joint training.** Joint optimization over both text-to-image (T2I) and image-to-text (I2T) objectives is essential. As shown in Tab. 4, joint training yields the highest GenEval score, outperforming both T2I-only and I2T-only variants. Notably, I2T-only causes GenEval to drop sharply from 61.6 to 28.3—more than a twofold decrease—while MS-COCO CIDEr remains nearly unchanged and VQAv2 declines only slightly. These results show that separating the objectives severely weakens cross-modal integration, underscoring the need for unified optimization to maintain strong multimodal coherence.

## 3.5 THE SCALABILITY OF MUDDIT

To demonstrate the scalability of our approach, we curate roughly 10 million image–text pairs from LAION-ART (Schuhmann et al., 2022), JourneyDB (Pan et al., 2023), CC12M (Changpinyo et al., 2021) and internal datasets. We filter out samples with an aesthetic score below 7, a height or width under 512 pixels, or an aspect ratio above 2. All images are re-captioned using Qwen2.5-VL 7B (Bai et al., 2023). We pretrain Muddit on this dataset with a batch size of 512 and a resolution of 1024, applying random masking to both image and text modalities. The image and text loss weights are set to 1.0 and 0.3, respectively. Training runs for 100K steps.

Table 6: Training hyperparameters across different training stages.

| Hyperparameters | Stage-I (Pre-training) | Stage-II (Instruction-tuning) |
|---|---|---|
| Learning Rate | $1.0 \times 10^{-4}$ | $1.0 \times 10^{-4}$ |
| LR Scheduler | Constant | Constant |
| Weight Decay | 0.01 | 0.01 |
| Max Gradient Norm | 10.0 | 10.0 |
| Optimizer | AdamW ($\beta_1 = 0.9, \ \beta_2 = 0.999$) | |
| Batch Size | 512 | 512 |
| Training Steps | 100K | 15K |
| Training GPUs | 16×H100 | 16×H100 |
| Gen. Resolution | 1024 | 1024 |
| Under. Resolution | 1024 | 1024 |

Table 7: Quantitative comparison with other unified models.

| Model | Params | Base model | Architecture | Data scale | Geneval w/ TTS | VQAv2 | MME | MMMU |
|---|---|---|---|---|---|---|---|---|
| Lumina-DiMOO (Xin et al., 2025b) | 8B | LLaDA | Discrete Diff. | 80M | 0.92 | – | 1534.2 | 58.6 |
| MMaDA ($512 \times 512$) (Yang et al., 2025) | 8B | LLaDA | Discrete Diff. | Unknown | 0.66 | 76.7 | 1410.7 | 30.2 |
| Show-O (512×512) (Xie et al., 2024) | 1.3B | Phi-1.5 | AR + Discrete Diff | 35M | – | 69.4 | 1097.2 | 27.4 |
| D-DiT (512×512) (Li et al., 2024b) | 2B | SD3-medium | Discrete Diff. + Diff. | 40M | – | 60.1 | 1124.7 | – |
| Muddit (512×512) | 1B | Meissonic | Discrete Diff. | 10M | 0.64 | 68.2 | 1107.4 | 27.6 |
| Muddit (1024×1024) | 1B | Meissonic | Discrete Diff | 16M | 0.67 | 70.2 | 1139.2 | 28.7 |

For instruction tuning, we collect about 6M samples from LLAVA-Instruct-150K (Liu et al., 2024e), ALLaVA LAION (Chen et al., 2024a), SA-1B (Kirillov et al., 2023), ART500K (Mao et al., 2017), ScienceQA (Lu et al., 2022), Chart2Text (Kantharaj et al., 2022), and VQAv2 (Antol et al., 2015). Muddit is then trained with a batch size of 512 at a resolution of 1024, with masking applied only to the answer text. We also add a 2M high-quality image dataset for high-quality fine-tuning. Further training configurations are provided in Tab. 6. All experiments are conducted on 16 H100 GPUs.

We evaluate the scaled Muddit model against other comparably sized unified models and state-of-the-art unified discrete diffusion models (Xin et al., 2025b; Yang et al., 2025), as shown in Tab. 7. Across established benchmarks, Muddit exhibits consistent improvements in both image generation and image understanding, empirically validating the scalability of our model. Furthermore, we compare Muddit with unified models of similar parameter sizes, all of which rely on hybrid architectures. Despite being trained on substantially less data, Muddit achieves superior performance.

We attribute this data efficiency to two key factors. First, our visual prior naturally maintains strong text-following capability for text-to-image generation, enabling robust alignment between image and text modalities. From the perspective of unified modeling, we prioritize cross-modal alignment over isolated single-modality ability, which allows Muddit to reach higher performance with less training data. Second, Muddit adopts a fully unified modeling paradigm: the model learns by predicting mask tokens based on context across all tasks (text-to-image and image-to-text). In contrast, hybrid architectures must simultaneously handle next-token prediction alongside velocity or mask prediction, and often introduce additional special tokens (*e.g.*, $< \text{soi} >$, $< \text{eoi} >$), which increases architectural complexity and hinders optimization.

## 4 CONCLUSION

In this work, we present Muddit, a unified generative framework that employs discrete diffusion to bridge text and image modalities. By unifying image and text generation within a single model, Muddit demonstrates strong performance across text-to-image, image-to-text, and VQA tasks. Notably, it matches or outperforms the capabilities of significantly larger autoregressive models, while enabling fast, parallel inference. Our results validate the effectiveness of discrete diffusion as a general-purpose modeling strategy and highlight its potential to serve as a scalable backbone for future multimodal systems that are equipped with strong visual priors.

**Acknowledgement.** This work is supported by the National Key Research and Development Program of China (No. 2023YFC3807600) and the National Natural Science Foundation of China under Grant No. 62320106007. In addition, this work is supported in part by NUS Start-up Grant A-0010106-00-00.

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

# APPENDIX

## APPENDIX OVERVIEW

This appendix provides additional discussions, results, and analyses to complement the main paper. It is organized as follows:

- **Related Work** (Sec. A): We review unified multimodal models for understanding and generation, with a focus on autoregressive and diffusion-based paradigms, as well as recent advances in masked image modeling.
- **Additional Qualitative Results** (Sec. B): We present extended visualizations for several tasks, including image captioning, text-to-image generation, visual question answering, and image-guided text editing.
- **Additional Experimental Results** (Sec. C): We present more experimental results.
- **Additional Ablation Studies** (Sec. D): We present extended ablation studies.
- **Inference Time Analysis** (Sec. E): We analyze inference efficiency by comparing autoregressive decoding with discrete diffusion, providing FLOPs complexity and speed benchmarks.
- **Generated Results Step by Step** (Sec. F): We illustrate the reverse discrete diffusion process in detail, showing intermediate decoding steps and examples of progressive generation.
- **Discussion** (Sec. G): We reflect on the limitations of our approach and its broader impacts, including potential applications and risks of misuse.
- **Use of Large Language Models** (Sec. H): We clarify the role of large language models during paper preparation.

## A    RELATED WORK

### A.1    UNIFIED MODELS FOR GENERATION AND UNDERSTANDING

The success of LLMs in language modeling has inspired efforts to extend unified generation to multimodal domains. However, the divergence between autoregressive and diffusion-based paradigms presents fundamental architectural trade-offs. Autoregressive models naturally handle language, and several works (Sun et al., 2023; Wang et al., 2024a; Tong et al., 2024; Ge et al., 2024; Dong et al., 2023; Chen et al., 2025b) extend this by connecting vision modules to LLMs via adapters or instruction tuning, with LLMs serving as planning modules that produce intermediate representations for image generation. While effective to some extent, these paradigms often exhibit limited interaction between text and image modalities and struggle with content consistency, particularly in image-to-image generation and complex instruction-based synthesis. To address these limitations, recent research explores unified generation models that integrate understanding and generation within a single architecture. We categorize these into four major paradigms (see Fig. 6):

**Fully Autoregressive**: Both text and image are tokenized into discrete sequences and modeled with an AR Transformer (Liu et al., 2024b; Team, 2024; Wu et al., 2024; Wang et al., 2024b; Chen et al., 2025d; Liu et al., 2024a; Guo et al., 2024; Zheng et al., 2024). These models achieve strong cross-modal generation but suffer from high latency due to sequential decoding.

**Text AR, Image Diffusion**: LLMs generate text tokens while image synthesis is delegated to pre-trained continuous diffusion backbones (Zhou et al., 2024; Zhao et al., 2024; Ma et al., 2024) or discrete diffusion (Xie et al., 2024). Though visually strong, these models are not truly unified, as they rely on separate architectures and token spaces.

**Image Diffusion, Text Discrete Diffusion**: Emerging models experiment with discrete diffusion for text and images (Li et al., 2024c), though many, like Dual-Diffusion (Li et al., 2024c), still use continuous diffusion for image synthesis, failing to realize true modality symmetry.

**Fully Discrete Diffusion**: Recent work like UniDisc (Swerdlow et al., 2025a) pioneers full-token discrete diffusion over shared Transformer backbones. These models support parallel sampling and native integration, but currently lag behind in generation fidelity and scale.

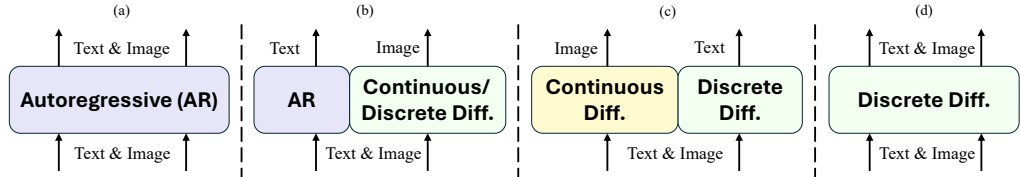

Figure 6: Four types of unified generative models. More details can be found in Sec. A.

Among these, the GPT-4o (OpenAI, 2025) model represents a significant advance as a unified multimodal generative system. However, its closed-source nature obscures critical architectural and training details, and its success may be largely attributable to scale rather than architectural novelty (Chen et al., 2025c).

## A.2 MASKED IMAGE MODELING

Masked Image Modeling (MIM) has emerged as a powerful self-supervised learning paradigm in computer vision (Bai et al., 2025a), drawing inspiration from the success of Masked Language Modeling (MLM) in NLP, notably BERT (Devlin, 2018). The fundamental principle of MIM involves obscuring portions of an image, which could be raw pixels (MAE (He et al., 2022)), latent patches of pixels, or even discrete latent tokens (BEiT (Bao et al., 2021), MaskGIT (Chang et al., 2022)), and training a model, typically an autoencoder, to predict or reconstruct this missing information by leveraging the context provided by the visible parts.

MaskGIT (Chang et al., 2022) introduced parallel decoding via iterative token refinement, inspiring discrete diffusion models. Recent work such as RandomAR (Fan et al., 2024) and MAR (Li et al., 2024a) formalize this as random-order or masked autoregressive generation, blending AR and MIM principles. The major conceptual difference between RandomAR/MAR and MaskGIT is in the scanning order at inference time.

This class of techniques forms the conceptual foundation of discrete diffusion over tokenized spaces (Shi et al., 2025) and plays a critical role in modern unified models. We will introduce discrete diffusion in the next section.

## A.3 RELATIONSHIP TO CONCURRENT WORK

Our main contribution is to demonstrate that a unified, *visual-prior* fully discrete diffusion model can be both effective and data-efficient for image understanding tasks, rather than just text-to-image generation tasks. Regarding the distinction from concurrent discrete diffusion works (Yang et al., 2025; Bai et al., 2026; Xin et al., 2025b;a), we think that unified models should allow for multiple design choices. Our goal is to demonstrate that a visual-first, fully discrete diffusion backbone can be a practical and competitive alternative to the more common "LLM-first" unified paradigm, and we believe this is a fundamental design choice.

Specifically, prior unified discrete diffusion models, such as UniDisc (Swerdlow et al., 2025b), are trained from scratch on multimodal data and therefore lack strong visual priors. As a result, they significantly underperform early diffusion baselines such as Stable Diffusion 1.5 (Rombach et al., 2022) and do not support visual question answering tasks (Antol et al., 2015). In contrast, Muddit is the first unified discrete diffusion model built on top of a pretrained high-resolution text-to-image backbone (Bai et al., 2025b), with a lightweight text decoder on top. This visual prior is not an implementation detail: it improves the scalability and generalization behavior of discrete diffusion through a visual backbone that is well aligned with text.

## B  ADDITIONAL QUALITATIVE RESULTS

**Image-to-text Generation.** We present more examples for image-to-text generation in Fig. 7.

**Text-to-image Generation.** We present more examples for text-to-image generation in Fig. 8.

**Visual Question Answering.** We present more examples for visual question answering in Fig. 9. Muddit reliably identifies fine-grained attributes (*e.g.*, "blonde" hair), object categories (*e.g.*, "beagle"), and physical affordances (*e.g.*, answering "No" to crossing at a red light). Notably, it also handles commonsense reasoning and spatial localization, such as inferring traffic legality or locating vehicles on the street.

**Image-guided text editing.** Zero-shot text-guided image editing performance is already verified and presented in Meissonic (Bai et al., 2025b). As the successor to Meissonic, we present Muddit's performance on the image-guided text editing task, where the model completes a masked sentence based on the input image. As shown in Fig. 10, given a partially masked caption and an image, Muddit fills in the blanks with semantically and visually grounded phrases.

## C  ADDITIONAL EXPERIMENTAL RESULTS

We provide a detailed breakdown of the MME benchmark results in Tab. 8. Muddit demonstrates strong performance in existence, color, and scene understanding, while also exhibiting solid reasoning capabilities.

Table 8: Detailed MME results.

| Category | Task | Score |
|---|---|---|
| | Existence | 135.00 |
| | Count | 78.33 |
| | Position | 53.33 |
| | Color | 140.00 |
| | Posters | 62.24 |
| Perception | Celebrity | 56.18 |
| | Scene | 107.25 |
| | Landmark | 94.50 |
| | Artwork | 76.00 |
| | OCR | 52.50 |
| | **Total** | **855.34** |
| | Commonsense Reasoning | 78.57 |
| | Numerical Calculation | 90.00 |
| Cognition | Text Translation | 57.89 |
| | Code Reasoning | 57.50 |
| | **Total** | **283.97** |

## D  ADDITIONAL ABLATION STUDIES

### D.1  ABLATION STUDY ON THE CFG FOR IMAGE-TO-TEXT GENERATION

As shown in Tab. 9, we report performance on MS-COCO captioning and VQAv2 benchmarks. Moderate CFG values (e.g., 1.5) yield the best results, while higher scales lead to degraded performance.

## E  INFERENCE TIME ANALYSIS

As shown in Fig. 13, autoregressive multimodal models are inherently limited by token-by-token decoding, which constrains their inference speed. Muddit overcomes this bottleneck with a parallel discrete diffusion decoder, reducing average latency to just 1.49 seconds, achieving a $4\times$ to $11\times$ speedup over competitive baselines ($4.2\times$ faster than Qwen-2.5-VL, $5.6\times$ than Show-o, $8.1\times$ than BLIP-2, and $10.9\times$ than LLaVA-1.6).

We also present detailed FLOPs comparison between Autoregressive and Discrete Diffusion.

Table 9: Ablation study on the effect of classifier-free guidance (CFG) scale.

| Dataset | CFG = 1 | CFG = 1.5 | CFG = 2 | CFG = 2.5 | CFG = 3 |
|---------|---------|-----------|---------|-----------|---------|
| MS-COCO | 57.2 | 59.9 | 58.2 | 51.3 | 47.2 |
| VQAv2 | 65.8 | 68.2 | 64.7 | 55.4 | 49.2 |

Table 10: Comparison of model efficiency across different resolutions and steps. We report throughput for both text-to-image generation (images per second) and image-to-text tasks (tokens per second). Muddit achieves the best overall balance, matching the highest text-to-image throughput while significantly outperforming others in image-to-text speed.

| Model | Image Res | Steps | Text-to-Image (img/s) | Image-to-Text (token/s) |
|-------|-----------|-------|-----------------------|-------------------------|
| Meissonic | 1024 | 32 | 0.23 | – |
| UniDisc | 512 | 32 | 0.89 | 79.36 |
| Monetico | 512 | 32 | 1.00 | – |
| D-DiT | 512 | 28 | 0.62 | 26.89 |
| Muddit | 512 | 32 | 1.00 | 99.98 |

**Autoregressive (AR) without KV Cache:**

- At step $t$, the model attends over $t$ previous tokens.
- Per-step attention FLOPs: $O(t^2 D)$.
- Total FLOPs:

$$\sum_{t=1}^{L} O(t^2 D) = O\left(D \sum_{t=1}^{L} t^2\right) = O\left(D \cdot \frac{L(L+1)(2L+1)}{6}\right) = O(L^3 D)$$

**Autoregressive (AR) with KV Cache:**

- At step $t$, Q is computed for 1 token, and attends to $t$ K/V keys.
- Per-step attention FLOPs: $O(tD)$.
- Total FLOPs:

$$\sum_{t=1}^{L} O(tD) = O\left(D \sum_{t=1}^{L} t\right) = O\left(D \cdot \frac{L(L+1)}{2}\right) = O(L^2 D)$$

**Discrete Diffusion:**

- Each step updates the full sequence (length $L$) in parallel.
- Per-step attention FLOPs: $O(L^2 D)$.
- Total FLOPs:

$$T \cdot O(L^2 D) = O(TL^2 D), \quad T \ll L$$

While discrete diffusion may appear less efficient than autoregressive (AR) models with KV caching in terms of theoretical FLOPs, it offers a significant advantage over AR without caching—achieving an L/T speedup by updating the full token sequence in parallel over T iterations. In practice, the higher degree of parallelism leads to competitive, and often faster, inference speed compared to AR models, especially when considering real-world GPU throughput. As KV cache techniques for discrete diffusion are rapidly evolving (Ma et al., 2025), we expect further acceleration in the near future, narrowing the theoretical speed gap even with KV-cache AR baselines.

In Tab. 10, we compared Muddit against other non-autoregressive models, running all tests on a single A800 80 GB GPU. Muddit demonstrated a clear advantage in both image and text generation.

## F    Generated Results Step by Step

Muddit frames text generation as reverse discrete diffusion over a fixed-length sequence of 77 token indices. At inference time, the model performs $16 \leq T \leq 32$ denoising steps, starting from a maximally entropic prior where every token is masked. At each step $t$, a parameter-shared transformer G predicts a categorical distribution over all positions in parallel, and a sampler S selects the next sequence:

$$\mathbf{x}_{t-1} = \mathtt{S}(\mathtt{G}(\mathbf{x}_t, \mathbf{c}, t), \mathbf{x}_t, t), \qquad t = T, \ldots, 1, \tag{12}$$

where $\mathbf{x}_t \in \mathbb{V}^{77}$ is the token sequence at step $t$, and c denotes conditioning inputs. The logits can be tempered or top-$k$ filtered before sampling each token independently. The resulting sequence $\mathbf{x}_{t-1}$ seeds the next step, enabling fast, parallel decoding without autoregressive constraints.

Because all positions are updated in parallel, Muddit preserves global syntactic and semantic structure throughout the reverse diffusion process—unlike left-to-right autoregressive models, which can only condition on past predictions. Empirically, as few as $16 \leq T \leq 32$ steps are sufficient to approximate the natural language distribution with high fidelity. Thus, Muddit unifies diffusion generation with parallel decoding, effectively overcoming the serial bottleneck that limits conventional autoregressive multimodal models.

We present two examples in Fig. 11 and 12.

## G    Discussion

### G.1    Limitations

While Muddit advances discrete diffusion for unified multimodal generation, it still has several limitations. First, due to its token-level discrete representation, the model may underperform continuous diffusion models in generating photorealistic or high-resolution images. Second, Muddit is initialized from a pretrained text-to-image foundation model, which offers strong visual priors but limits its capacity for rich text understanding and generation compared to the latest large language models. This makes it less suitable for tasks that require long-form understanding and generation or deep linguistic reasoning.

### G.2    Broader Impacts

Muddit explores a new paradigm in multimodal generation by leveraging a strong visual prior as the backbone, in contrast to the prevailing trend of scaling large language models. This offers a complementary path toward efficient, grounded multimodal generation, particularly in vision-centric applications. The model's ability to generate aligned visual and textual outputs in a fast, parallel manner could benefit downstream tasks, especially in completion-based scenarios such as masked captioning, image editing, and code implementation. However, as with all generative models, there remains a risk of misuse in synthetic content creation.

## H    The Use of Large Language Models

During the preparation of this paper, large language models were used only for language polishing and minor editing. All research ideas, methods, and experimental results were carried out entirely by the human authors.

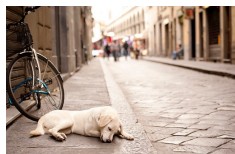

**Assistant:** The image depicts a charming street scene with a cobblestone pavement and tall buildings. A white dog is resting. The dog is lying down near the bicycle on the street. To the left of the street, there is an old bicycle parked to the side of the street.

**Assistant:** The image is a black - and - white photograph featuring three individuals sitting together on a bench. They appear to be in an urban setting, as indicated by the trees and buildings in the background. The individuals are dressed in police uniforms, with one person looking behind the first individual from the second individual.

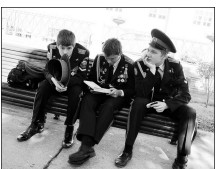

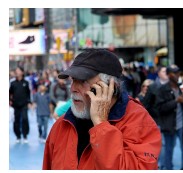

**Assistant:** The image depicts an elderly man standing on a city street, wearing a red jacket and a black cap talking on cell phone to his ear. The background shows a crowd of people walking around, some of whom are handbags and others visible in the distance. The crowd appears to be the middle of a busy area.

**Assistant:** The image depicts a close - up of a large, round pizza placed on a wooden table. The pizza is topped with cheese, and there are several pieces of basil leaves. In the background, there are other people sitting around.

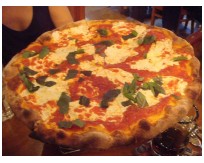

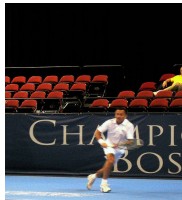

**Assistant:** The image depicts a tennis match on a blue court. the tennis match is set against a backdrop of a blue court. The player, wearing a white shirt and white shorts, is in mid action and holding a tennis racket, possibly preparing for a shot. the court has a row of seats.

**Assistant:** The image depicts a vibrant and colorful arrangement of umbrellas hanging from the branches of trees. The umbrellas come in color and shape, including pink, red, and yellow, blue, and purple. They are arranged in a circular pattern across the umbrellas, creating a visually appealing display. The background features a mix of green trees and foliage, suggesting an outdoor.

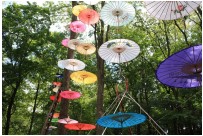

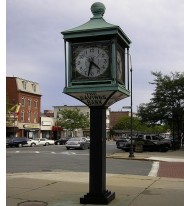

**Assistant:** The image depicts a bustling urban street scene with several notable elements. In the foreground, there is a tall green clock tower with a clock face and a pole, which stands prominently on the street. The clock has a clock face on it, indicating the time. In the background, there are several parked cars.

**Assistant:** The image depicts a person surfing on a surfboard on a large wave. The surfer is in mid-air, with his arms bent, navigating the balance of the wave. The background shows a clear blue sky, adding depth to the overall scene. The surfer seems to be in the middle of the the wave.

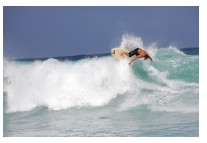

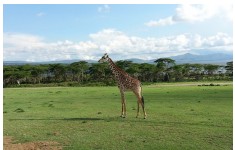

**Assistant:** The image depicts a giraffe standing on a grassy field in a rural outdoor setting. The background features a clear blue sky with some scattered clouds, and there are trees in the distant distance. The giraffe appears to be looking towards a distant horizon.

**Assistant:** The image depicts a stop sign in front of a grassy field with majestic mountains in the background. The sky is clear with a scattered clouds, and snow-capped mountains rise in the background. The scene appears to be in a rural setting.

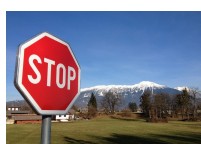

Figure 7: Image-to-text generated results.

**Human**: An editorial fashion photo portrait of a striking avant-garde model with bubble details. Iridescent art and pop surrealism influence the image's ethereal feel. Pastelaesthetic hues of soft pink, tangerine, and green create a dreamy backdrop. Seamless pink background, studio lighting emphasizes the model's silhouette against the soft pastel palette.

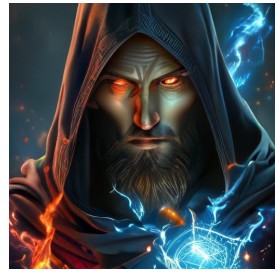

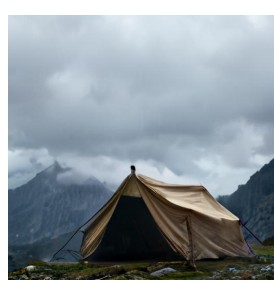

**Human**: Dark, heavy rainclouds gather over a rocky mountain range, obscuring the peaks. A weathered canvas tent flaps open in the wind, its silhouette stark against the gray backdrop. Water pours down in sheets, blurring the image and creating a misty effect.

**Human**: A quaint small town nestled amidst snow-capped hills, bathed in the soft, golden hues of dusk. The sky is a tapestry of twilight blue and orange, casting long shadows across cobblestone streets as frosted trees stand bare against the fading light. A sense of quiet peace emanates from the scene.

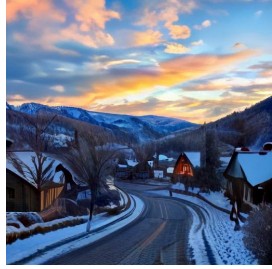

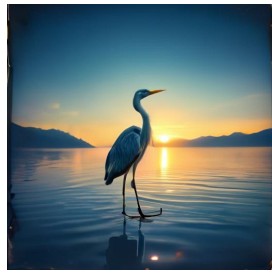

**Human**: An old cyanotype photograph capturing a serene lake at twilight. A majestic heron stands in the tranquil waters, its long legs poised, as the last golden rays of sunlight paint the sky with hues of blue and purple. The distant mountain range glows warmly with the soft, crepuscular light. Sharp focus on the heron and water reflection. Wide-angle lens captures a panoramic scene.

**Human**: A vibrant Japanese garden, inspired by Van Gogh's swirling brushstrokes. Vivid reds, yellows, blues, and greens dominate the scene. Cherry blossoms bloom in full splendor against a backdrop of ancient stone lanterns, rendered with lush foliage and dappled sunlight. The effect is heightened by 4K resolution and cinematic depth.

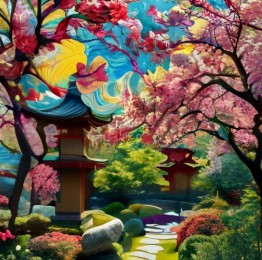

Figure 8: Text-to-image generation results.

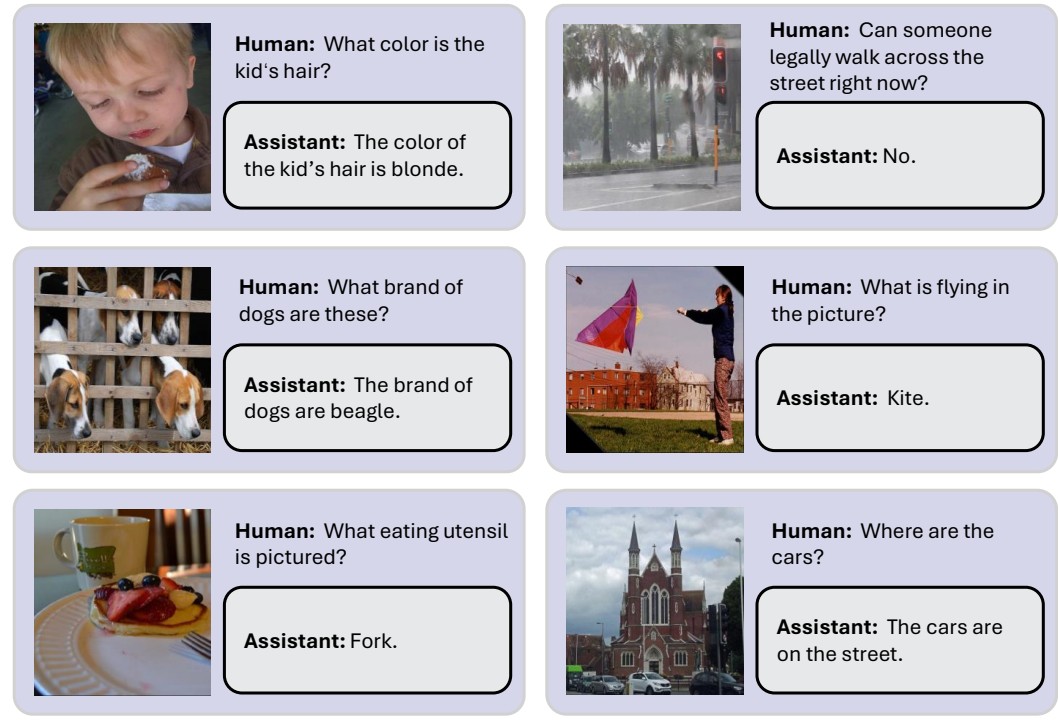

Figure 9: Visual question answering results.

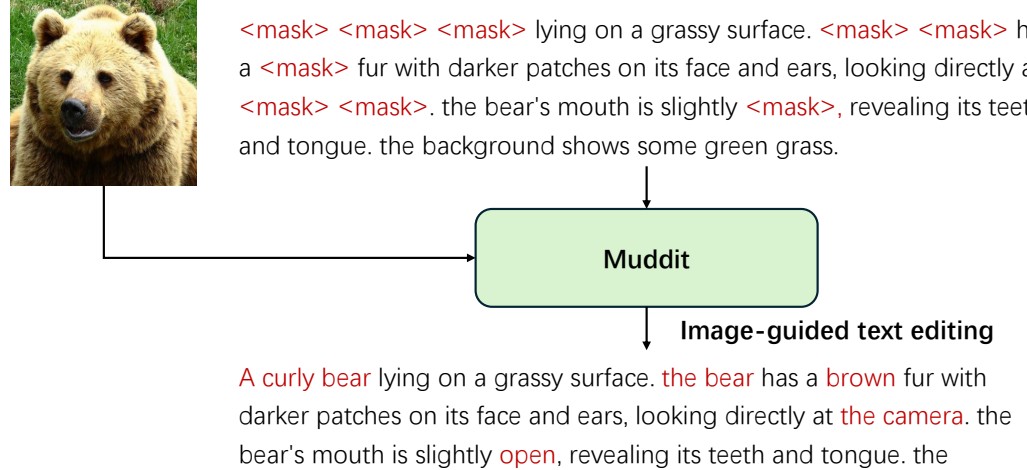

Figure 10: Image-guided text editing results.

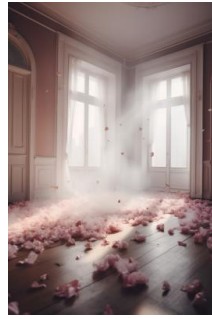

**User:** Please describe this image.

**T=1**

<mask> <mask> <mask> <mask> <mask> <mask> <mask> <mask> <mask> <mask> <mask> <mask> <mask> <mask> <mask> <mask> <mask> <mask> <mask> <mask> <mask> <mask> <mask> <mask> <mask> <mask> <mask> <mask> <mask> <mask> <mask> <mask> <mask> <mask> <mask> <mask> <mask> <mask> <mask> <mask> <mask> <mask> <mask> <mask> <mask> <mask> <mask> <mask> <mask> <mask> <mask> <mask> <mask> <mask> <mask> <mask> <mask> <mask> <mask> <mask> <mask> <mask> <mask> <mask> <mask> <mask> <mask> <mask> <mask> <mask> <mask> <mask> <mask> <mask> <mask> <mask> <mask> <mask> <mask>

<mask> <mask> depicts <mask> <mask> and whimsical <mask> <mask> <mask> <mask> <mask> <mask> <mask> <mask> <mask>, <mask> <mask> <mask> <mask> <mask> <mask> <mask>. <mask> <mask> <mask> <mask> <mask> <mask> <mask> <mask> <mask> <mask> <mask> <mask> <mask> <mask> <mask> <mask> <mask> <mask> <mask> <mask> <mask> <mask> <mask> <mask> <mask> <mask> <mask> <mask> <mask> <mask> <mask> <mask> <mask> <mask> <mask> <mask> <mask> <mask> <mask> <mask> <mask> <mask> <mask> <mask>

The <mask> depicts a <mask> and whimsical <mask> <mask> <mask> <mask> <mask> <mask> <mask> <mask> <mask>, <mask> <mask> <mask> <mask> <mask> <mask> <mask>. the <mask> <mask> <mask> <mask> <mask> <mask> <mask> <mask> <mask> <mask> <mask> <mask> <mask> <mask> <mask> <mask> <mask> <mask> <mask> <mask> <mask> <mask> <mask> <mask> <mask> <mask> <mask> <mask> <mask> <mask> <mask> <mask> <mask> <mask> <mask> <mask> <mask>, <mask> <mask> <mask> <mask> <mask> <mask>

The image depicts a surreal and whimsical scene in what appears to be a <mask> <mask>, possibly <mask> <mask> or a dining room. the floor is covered with <mask> <mask> <mask> <mask> <mask> <mask> <mask> <mask> <mask> <mask> <mask> <mask> <mask> <mask> <mask> <mask> <mask> <mask> <mask> <mask> <mask> <mask> <mask> <mask> <mask> <mask> <mask> <mask> <mask> <mask>, <mask> <mask> <mask> <mask> <mask>

The image depicts a surreal and whimsical scene in what appears to be a domestic setting, possibly a room or a dining room. the floor is covered with numerous pink flowers, adding a touch of <mask> <mask> <mask> <mask> <mask> <mask> <mask> <mask> <mask>, <mask> to the <mask> <mask> <mask> <mask> <mask> <mask> <mask>

The image depicts a surreal and whimsical scene in what appears to be a domestic setting, possibly a room or a dining room. the floor is covered with numerous pink flowers, adding a touch of <mask>. the petals are scattered throughout the room, adding to the dreamlike quality of the scene.

The image depicts a surreal and whimsical scene in what appears to be a domestic setting, possibly a room or a dining room. the floor is covered with numerous pink flowers, adding a touch of color. the petals are scattered throughout the room, adding to the dreamlike quality of the scene.

The image depicts a surreal and whimsical scene in what appears to be a domestic setting, possibly a room or a dining room. the floor is covered with numerous pink flowers, adding a touch of color. the petals are scattered throughout the room, adding to the dreamlike quality of the scene.

**T=0**

Figure 11: Image-to-text generated results in each step.

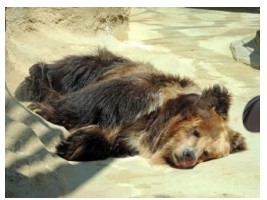

**User:** Please describe this image.

**T=1**

<mask> <mask> <mask> <mask> <mask> <mask> <mask> <mask> <mask> <mask> <mask> <mask> <mask> <mask> <mask> <mask> <mask> <mask> <mask> <mask> <mask> <mask> <mask> <mask> <mask> <mask> <mask> <mask> <mask> <mask> <mask> <mask> <mask> <mask> <mask> <mask> <mask> <mask> <mask> <mask> <mask> <mask> <mask> <mask> <mask> <mask> <mask> <mask> <mask> <mask> <mask> <mask> <mask> <mask> <mask> <mask> <mask> <mask> <mask> <mask> <mask> <mask> <mask> <mask> <mask> <mask> <mask> <mask> <mask>

<mask> image <mask> <mask> bear <mask> <mask> <mask> <mask> <mask> <mask> <mask> <mask> <mask> <mask> <mask> <mask> <mask> <mask> <mask> <mask> <mask> <mask> <mask> <mask> <mask> <mask> <mask> <mask> <mask> <mask> <mask> <mask> <mask> <mask> <mask> <mask> <mask> <mask> <mask> <mask> <mask> <mask> <mask> <mask> <mask> <mask> <mask> <mask> <mask> <mask> <mask> <mask> <mask> <mask> <mask> <mask> <mask> <mask> <mask> <mask> <mask> <mask> <mask> <mask> <mask> <mask>

<mask> image depicts <mask> bear lying <mask> its <mask> <mask> <mask> <mask> <mask> <mask> <mask> <mask> <mask> <mask> <mask> <mask> <mask> <mask> <mask> <mask> <mask> <mask> <mask> <mask> <mask> <mask> <mask> <mask> <mask> <mask> <mask> <mask> be <mask> <mask> <mask> <mask> <mask> <mask> <mask> <mask> <mask> <mask> <mask> <mask> <mask> <mask> <mask> <mask> <mask> <mask> <mask> <mask> <mask> <mask> <mask> <mask> <mask> <mask> <mask> <mask>.

<mask> image depicts <mask> bear lying <mask> its <mask> <mask> a <mask> <mask> <mask> <mask> <mask> <mask> <mask> <mask> <mask> <mask> <mask> <mask> <mask> <mask> <mask> <mask> <mask> <mask> <mask> <mask> <mask> <mask> <mask> <mask> <mask> appears to be resting <mask> <mask> <mask> <mask> <mask> <mask> <mask> <mask> <mask> <mask> <mask> <mask> <mask> <mask> <mask> <mask> <mask> <mask> <mask> <mask> <mask> <mask> <mask> <mask> <mask> or <mask>.

The image depicts a bear lying <mask> its back <mask> a <mask> surface <mask> <mask> <mask> <mask> <mask> <mask> <mask> <mask> <mask> <mask> <mask> <mask> <mask> <mask> <mask> <mask>. the bear appears to be sleeping <mask> with its head <mask> on <mask> <mask> surface <mask> <mask> <mask> <mask> <mask> <mask> <mask> <mask> <mask> <mask> <mask> <mask> <mask> or <mask>.

the image depicts a bear lying on its back on a <mask> surface. the <mask> is <mask> <mask> <mask> <mask> <mask> <mask> <mask> <mask> <mask> <mask> <mask> <mask> <mask>. the bear appears to be sleeping. the <mask> suggests <mask> the <mask> <mask> <mask>, possibly near a <mask> or <mask>.

The image depicts a bear lying on its back on a concrete surface. the bear is lying down, with <mask> head <mask> <mask> its head resting <mask> <mask> <mask>. the bear appears to be sleeping. the setting suggests that the scene is outdoors, possibly near a park or <mask>.

The image depicts a bear lying on its back on a concrete surface. the bear is lying down, with its head resting on the surface. the bear appears to be sleeping. the setting suggests that the scene is outdoors, possibly near a park or garden.

**T=0**

Figure 12: Image-to-text generated results in each step.

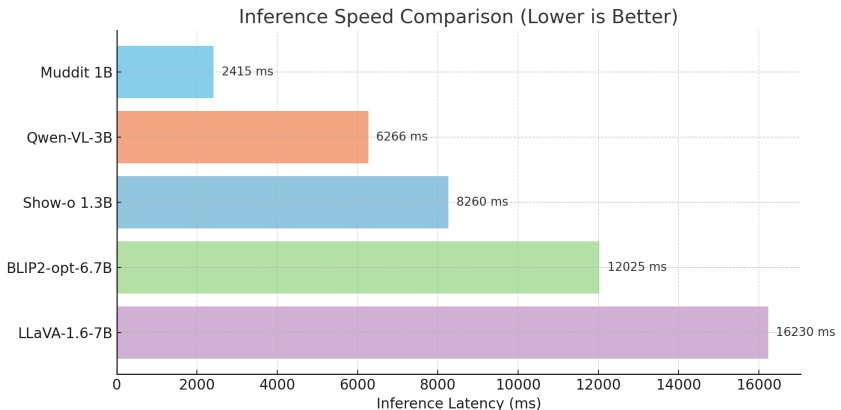

Figure 13: Inference speed comparison. We use 32 inference steps for Muddit and fix the sequence length to 77 across all models.

