# OpenReview forum: "Beyond Text-to-Image: Liberating Generation with a Unified Discrete Diffusion Model"
_ICLR.cc/2026/Conference — ICLR 2026 Poster_

### Official Review · Reviewer_G1Hq · 2025-10-31

**Soundness:** 3
**Presentation:** 3
**Contribution:** 3
**Rating:** 4
**Confidence:** 3

**Summary:**

This paper presents Muddit, a unified generative framework that applies discrete diffusion for both text and image modalities. The model leverages a pretrained visual prior to enable fast and parallel generation for variety of tasks. Experimental results show that Muddit achieves competitive or superior performance compared to larger autoregressive models.

**Strengths:**

- The paper is well-written and clearly structured, with detailed experiments across multiple multimodal tasks.
- It demonstrates that a purely discrete diffusion framework can achieve results comparable to autoregressive baselines while offering faster inference.
- The approach shows practical value by unifying image and text generation within a single framework.

**Weaknesses:**

Although overall quality of the paper and the proposed methods are good, I have two major concerns.
- It is not entirely clear why the unified use of discrete diffusion across modalities is fundamentally advantageous. The paper does not convincingly explain what conceptual or empirical benefits this fully discrete (with discrete diffusion) setup provides beyond architectural neatness. For example, from Table 1 and 2, models employing hybrid or alternative architectures perform better in some cases. A more detailed justification or analysis of why full discretization is beneficial would strengthen the contribution.
- The paper does not sufficiently distinguish its method from prior works that already employ discrete diffusion for both text and image. The differences seem limited to implementation choices and pre-trained priors rather than introducing a fundamentally new formulation. Clearer explanation showing how Muddit’s design overcomes the limitations of previous discrete diffusion models would make the contribution more substantial.

**Questions:**

See the weaknesses two

---

> ### Author Response · Authors · 2025-11-26
> **(1/2) Response to Reviewer G1Hq**
>
> > W1. It is not entirely clear why the unified use of discrete diffusion across modalities is fundamentally advantageous. The paper does not convincingly explain what conceptual or empirical benefits this fully discrete (with discrete diffusion) setup provides beyond architectural neatness. For example, from Table 1 and 2, models employing hybrid or alternative architectures perform better in some cases. A more detailed justification or analysis of why full discretization is beneficial would strengthen the contribution.
>
> We appreciate the reviewer’s question and provide further details on the advantages of a fully discrete diffusion architecture over hybrid designs. Conceptually, our goal is to show that a single discrete diffusion principle shared across modalities yields a simpler and more coherent unified generative framework than current pure AR or AR–diffusion hybrids.
>
> First, a fully discrete formulation allows us to treat text and image tokens under the same corruption–reconstruction process and sampler. In Muddit, all tasks, including text-to-image, image captioning, and VQA, use, the identical discrete diffusion, the same timestep schedule, and the same guidance operator; only the conditioning signal (text, image, or text+image) changes. This symmetry enables truly unified operations such as joint infilling and cross-modal editing within a single backbone and sampling procedure, instead of stitching together a language model and an image diffusion model with task-specific glue as in many hybrid AR–diffusion systems. It also makes the framework naturally extensible to additional discrete modalities (e.g., audio tokens, 3D tokens) without redesigning the generative principle.
>
> Second, the fully discrete setup simplifies the learning objective and improves training efficiency. Muddit always performs bidirectional masked-token prediction conditioned on the available context (text, image, or both) for all tasks. This is closer to masked language modeling or MaskGIT-style image modeling, where each update can leverage information from both “left” and “right” contexts, and all tokens are updated in parallel at each diffusion step. In contrast, hybrid unified architectures must simultaneously manage autoregressive next-token prediction for text and velocity or mask prediction for images, often with additional special tokens (e.g., [soi], [eoi], [sot], [eot]) and task-specific routing. This mixture of objectives complicates optimization and forces the model to learn two different generation paradigms at once. Empirically, we observe that the purely masked, bidirectional objective converges faster for a given number of updates and leads to better data efficiency in the unified setting.
>
> Empirically, as summarized in the table below, we compare Muddit against unified models with similar parameter sizes that all employ hybrid architectures. Despite using substantially less training data, Muddit achieves superior performance on representative multimodal understanding benchmarks (and we also include additional scaling experiments in Table 6 and Table 7 in the revised manuscript).
>
> | Model | Params | Base model | Architecture | Data scale | VQAv2 | MME | MMMU |
> | :--- | :--- | :--- | :--- | :--- | :--- | :--- | :--- |
> | UniDisc (512$\times$512) | 1.4B | - | Discrete Diff | 280M | - | - | - |
> | Show-O (512$\times$512) | 1.3B | Phi-1.5 | AR + Discrete Diff | 35M | 69.4 | 1097.2 | 27.4 |
> | D-DiT (512$\times$512) | 2B | SD3-medium | Discrete Diff + Diff | 40M | 60.1 | 1124.7 | - |
> | **Muddit (1024$\times$1024)** | **1B** | **Meissonic** | **Discrete Diff** | **16M** | **70.2** | **1139.2** | **28.7** |
>
> Compared to Show-O and D-DiT, a 1B-parameter Muddit trained on less than half the data attains higher VQAv2 and MME scores and a stronger MMMU score, while retaining non-autoregressive, parallel inference for both text and images. These results indicate that full discrete diffusion, when combined with strong visual priors and a unified masked-token objective, is not merely architecturally neat but a practically viable alternative that offers a better balance between quality, efficiency, and architectural simplicity than existing hybrid unified models. In the revised version, we have clarified these conceptual and empirical advantages and include the above comparison in the revised manuscript.

---

> ### Author Response · Authors · 2025-11-26
> **(2/2) Response to Reviewer G1Hq**
>
> > W2. The paper does not sufficiently distinguish its method from prior works that already employ discrete diffusion for both text and image. The differences seem limited to implementation choices and pre-trained priors rather than introducing a fundamentally new formulation. Clearer explanation showing how Muddit’s design overcomes the limitations of previous discrete diffusion models would make the contribution more substantial.
>
> Our main contribution is to demonstrate that a unified, visual-prior fully discrete diffusion model can be both effective and data-efficient for image understanding tasks, rather than just text-to-image generation tasks. Regarding the distinction from prior discrete diffusion works, we think that unified models should allow for multiple design choices. Our goal is to demonstrate that a visual-first, fully discrete diffusion backbone can be a practical and competitive alternative to the more common “LLM-first” unified paradigm, and we believe this is a fundamental design choice.
>
> Specifically, prior unified discrete diffusion models, such as UniDisc [1], are trained from scratch on multimodal data and therefore lack strong visual priors. As a result, they significantly underperform early diffusion baselines such as Stable Diffusion 1.5 and do not support vision question answering tasks. In contrast, Muddit is the first unified discrete diffusion model built on top of a pretrained high-resolution text-to-image backbone (Meissonic), with a lightweight text decoder on top. This visual prior is not an implementation detail: it improves the scalability and generalization behavior of discrete diffusion through a text well-aligned visual backbone.
>
> Empirically, this difference manifests in both quality and data efficiency. With only 1B parameters, Muddit improves GenEval overall accuracy from 0.42 (UniDisc) to 0.61, and surpasses Monetico and Meissonic on text-to-image benchmarks. At the same time, the same 1B discrete diffusion model achieves competitive performance on MS-COCO captioning, VQAv2, MME, MMBench, GQA, and MMMU, closing much of the gap to significantly larger language-prior unified models, like 8B MMaDA [2]. We present detailed comparisons in Table 6 and Table 7 in the revised manuscript.
>
> Finally, to the best of our knowledge (excluding concurrent work), Muddit is the first fully discrete diffusion model that supports text-to-image, image captioning, and visual question answering within a single backbone and a single sampling procedure.
>
> * [1] UniDisc: Unified Multimodal Discrete Diffusion, https://arxiv.org/abs/2503.20853
> * [2] MMaDA: Multimodal Large Diffusion Language Models, https://arxiv.org/abs/2505.15809

---

### Official Review · Reviewer_aAn6 · 2025-11-01

**Soundness:** 4
**Presentation:** 4
**Contribution:** 4
**Rating:** 8
**Confidence:** 3

**Summary:**

This paper presents Muddit, which is a unified discrete diffusion model for text-to-image, image-to-text, and VQA tasks. Achieving competitive performance with existing AR or diffusion or AR/diffusion mixture models, Muddit, with a relatively small model size (1B), demonstrates the potential of a purely discrete-diffusion-based model as a scalable and effective backbone for unified generation.

**Strengths:**

- Muddit is built upon a visual prior rather than a language prior, which offers a new path in bridging vision and language into a unified model.
- Muddit achieved comparable or superior performance with a relatively small model size.

**Weaknesses:**

The performance of Muddit is comparable to or slightly better than Show-o (size 1.3B) in both Table 1: Text-to-image generation performance and Table 2: Image captioning and VQA benchmarks.

**Questions:**

1. I didn’t find explicit evidence in the paper to support the scalability potential of Muddit. Could the authors further elaborate on the meaning of "scalable" in the claim: "highlight Muddit's potential to serve as a scalable backbone for future multimodal systems" at the end of the conclusion section (and the abstract section)?
2. Some of the evaluation datasets are split into specific tasks (i.e. MME, which contains multiple categories, such as Existence, Count, Position, Color, OCR, Poster, Celebrity, etc). It would be interesting to also include detailed performance under different task categories.

---

> ### Author Response · Authors · 2025-11-26
> **Response to Reviewer aAn6**
>
> > W1 and Q1. The scalability potential of Muddit.
>
> We appreciate the suggestion. We curated an additional 6M samples, including 2M image–caption pairs and 4M VQA pairs spanning diverse domains such as daily activities, animals, art, science, and reasoning. Muddit was trained on this dataset for one epoch with a batch size of 512 at a resolution of 1024. Evaluation on established benchmarks shows that Muddit consistently outperforms comparable-sized unified models in both image generation and image understanding tasks, empirically validating the scalability of our approach.
>
> | Model | Params | Data scale | Geneval | VQAv2 | MME | MMMU |
> | :--- | :--- | :--- | :--- | :--- | :--- | :--- |
> | Muddit ($512 \times 512$) | 1B | 10M | 0.61 | 68.2 | 1107.4 | 27.6 |
> | **Muddit ($1024 \times 1024$)** | **1B** | **16M** | **0.62** | **70.2** | **1139.2** | **28.7** |
>
> > Q2. Some of the evaluation datasets are split into specific tasks (i.e. MME, which contains multiple categories, such as Existence, Count, Position, Color, OCR, Poster, Celebrity, etc). It would be interesting to also include detailed performance under different task categories.
>
> Thank you for the suggestion. We have included the per-category scores of the scaled Muddit model on the MME benchmark and added this table in Appendix C.
>
> | Category | Task | Score |
> | :--- | :--- | :--- |
> | **Perception** | Existence | 135 |
> | | Count | 78.33 |
> | | Position | 53.33 |
> | | Color | 140 |
> | | Posters | 62.24 |
> | | Celebrity | 56.18 |
> | | Scene | 107.25 |
> | | Landmark | 94.5 |
> | | Artwork | 76 |
> | | OCR | 52.5 |
> | | **Total** | **855.34** |
> | **Cognition** | Commonsense Reasoning | 78.57 |
> | | Numerical Calculation | 90 |
> | | Text Translation | 57.89 |
> | | Code Reasoning | 57.5 |
> | | **Total** | **283.97** |

---

### Official Review · Reviewer_TaRs · 2025-11-01

**Soundness:** 3
**Presentation:** 3
**Contribution:** 2
**Rating:** 4
**Confidence:** 4

**Summary:**

This paper presents Muddit, a unified generative model for text and images that challenges the dominance of autoregressive (AR) frameworks. The authors argue that AR models are slow and that existing unified diffusion models are weak because they are trained from scratch, lacking strong priors. Muddit's core contribution is a "visual-first" approach. It uses a single discrete diffusion (MaskGIT-style) transformer backbone that is initialized with the weights of a powerful, pretrained text-to-image model (Meissonic). A new lightweight text decoder is then added, and the entire 1B parameter model is jointly trained (via pretraining and instruction tuning) to handle text-to-image, image captioning, and VQA tasks using a single, parallel decoding process.

**Strengths:**

1. The "visual-first" strategy is a novel and compelling way to build a unified model. It cleverly solves the "weak prior" problem of from-scratch diffusion models by inheriting the strong visual capabilities of Meissonic, while using a single, symmetric discrete diffusion objective for all tasks.
2. The model's primary strength is its efficiency. By replacing sequential AR decoding with parallel discrete diffusion, the paper demonstrates a significant inference speedup over prominent AR models (Fig. 13 and Tab. 7), a practical advantage for real-world applications.
3. By building on a SOTA T2I model, Muddit's image generation quality is excellent for its size. It achieves a 0.61 on GenEval, making it competitive with much larger models and even SOTA diffusion-only models like SD3 (0.62).

**Weaknesses:**

1. The model's key strength is also its greatest weakness. By building on a visual backbone, the model's language and reasoning capabilities are fundamentally limited. This is evident in Table 2, where Muddit's VQAv2 (68.2) scores are significantly lower than LLM-based AR models like LLaVA-Next (82.8) or discrete diffusion unified model like Show-o (69.4).
2. The model is only 1B parameters. The model capacity and visual-first training methodology bring high efficiency but make it lack far behind the leading unified discrete diffusion models like MMaDA (Yang et al., 2025) and Lumina-DiMOO (Xin et al., 2025). As a unified discrete diffusion model, the author should compare to SOTA method like MMaDA in the paper.

**Questions:**

1. The text decoder is a "lightweight linear head" on a visual backbone. Is this the primary bottleneck for the I2T/VQA tasks? Can the authors elaborate on why a more sophisticated text decoder (e.g., a small size Transformer) was not used?
2. Does the author believe this "visual-first" architecture can scale to 7B+ parameters and eventually close the gap on language-heavy tasks, or is it primarily intended for vision-centric tasks where language is secondary?

---

> ### Author Response · Authors · 2025-11-26
> **(1/2) Response to Reviewer TaRs**
>
> > W1. The model's key strength is also its greatest weakness. By building on a visual backbone, the model's language and reasoning capabilities are fundamentally limited. This is evident in Table 2, where Muddit's VQAv2 (68.2) scores are significantly lower than LLM-based AR models like LLaVA-Next (82.8) or discrete diffusion unified model like Show-o (69.4).
>
> We agree with the reviewer that Muddit does not match the language and reasoning capabilities of large LLM-based AR models such as LLaVA-Next on VQAv2. This is expected given our design and resource regime: Muddit uses a 1B-parameter visual-first backbone, no large-scale language pretraining, and a lightweight text decoder, whereas LLaVA-Next builds on a much larger LLM and substantially more text-only data.
>
> Our goal in this work is therefore not to compete directly with 7B–70B LLM-first systems on language-heavy benchmarks, but to investigate whether a visual-first, fully discrete diffusion backbone can support competitive vision-centric multimodal reasoning while retaining high-quality image generation and efficient, parallel inference within a single unified framework. In this regime, the interaction between image and text representations and the ability to perform grounded reasoning are more critical than matching the absolute language strength of a large LLM.
>
> Empirically, as summarized below, when comparing to unified models of similar or larger size that employ hybrid architectures, Muddit achieves stronger VQAv2, MME, and MMMU scores despite using significantly less multimodal data:
>
> | Model | Params | Base model | Architecture | Data scale | VQAv2 | MME | MMMU |
> | :--- | :--- | :--- | :--- | :--- | :--- | :--- | :--- |
> | Show-O (512$\times$512) | 1.3B | Phi-1.5 | AR + Discrete Diff | 35M | 69.4 | 1097.2 | 27.4 |
> | D-DiT (512$\times$512) | 2B | SD3-medium | Discrete Diff + Diff | 40M | 60.1 | 1124.7 | - |
> | **Muddit (1024$\times$1024)** | **1B** | **Meissonic** | **Discrete Diff** | **16M** | **70.2** | **1139.2** | **28.7** |
>
> These results suggest that visual priors do not inherently limit image understanding or grounded reasoning; under a 1B parameter budget and modest data scale, a visual-first fully discrete diffusion model can already achieve competitive cross-modal performance against hybrid unified models. We have clarified this intended scope and positioning in the revised manuscript, Appendix A.3.
>
> > W2. The model is only 1B parameters. The model capacity and visual-first training methodology bring high efficiency but make it lack far behind the leading unified discrete diffusion models like MMaDA (Yang et al., 2025) and Lumina-DiMOO (Xin et al., 2025). As a unified discrete diffusion model, the author should compare to SOTA method like MMaDA in the paper.
>
> We appreciate the suggestion to compare Muddit against recent SOTA unified discrete diffusion models such as MMaDA and Lumina-DiMOO. We fully agree that our 1B-parameter, visual-first configuration cannot match the raw capacity of these 8B models, which are designed to operate in a very different parameter and data regime.
>
> Our intent in this work is to explore a complementary operating point: unified discrete diffusion models whose parameter budget is comparable to standard text-to-image backbones (e.g., SDXL-like models), rather than to large LLMs. To make this positioning clearer, we summarize below the model sizes, data scales, and representative metrics reported in the literature:
>
> | Model | Params | Base model | Data scale | GenEval w/ TTS | VQAv2 | MME | MMMU |
> | :--- | :--- | :--- | :--- | :--- | :--- | :--- | :--- |
> | Lumina-DiMOO | 8B | LLaDA | 80M | 0.92 | - | 1534.2 | 58.6 |
> | MMaDA ($512 \times 512$) | 8B | LLaDA | Unknown | 0.66 | 76.7 | 1410.7 | 30.2 |
> | Show-O ($512 \times 512$) | 1.3B | Phi-1.5 | 35M | - | 69.4 | 1097.2 | 27.4 |
> | **Muddit ($1024 \times 1024$)** | **1B** | **Meissonic** | **16M** | **0.67** | **70.2** | **1139.2** | **28.7** |
>
> As the table shows, Muddit sits between small LLM-based unified models (Show-O, D-DiT) and large LLM-based unified models (MMaDA, Lumina-DiMOO) in terms of capacity and data, while achieving competitive or better performance than models of similar size and significantly lower cost. We have also expanded the related work section in Appendix A.3 to more explicitly discuss MMaDA and Lumina-DiMOO and to emphasize that Muddit targets a lower-resource, vision-centric regime that is complementary to these large-scale unified systems.

---

> ### Author Response · Authors · 2025-11-26
> **(2/2) Response to Reviewer TaRs**
>
> > Q1. The text decoder is a "lightweight linear head" on a visual backbone. Is this the primary bottleneck for the I2T/VQA tasks? Can the authors elaborate on why a more sophisticated text decoder (e.g., a small size Transformer) was not used?
>
> To validate the efficacy of our text decoder, we conducted an experiment on the base model -- Meissonic. We found that a simple linear head could accurately reconstruct input text from the model's output embeddings, even when the base model was frozen, and only the head was trainable. Specifically, we train the lightweight linear head on the MS-COCO training set, fix the timestep to 0, which means no mask is applied. We compute the loss on text only. We evaluate the linear head on MS-COCO validation set, and try to reconstruct the input caption. We calculate the CIDEr score and show it in the table below. The result indicates that a lightweight linear head is sufficient for decoding, provided the text embeddings are accurately predicted. Consequently, we prioritized our architectural design and resources on the MM-DiT backbone to optimize multimodal processing.
>
> | Params | Text decoder | Data scale | MS-COCO CIDEr |
> | :--- | :--- | :--- | :--- |
> | Meissonic 1B | Linear | 118K | 83.4 |
>
> > Q2. Does the author believe this "visual-first" architecture can scale to 7B+ parameters and eventually close the gap on language-heavy tasks, or is it primarily intended for vision-centric tasks where language is secondary?
>
> In the design of Muddit, our model was indeed primarily intended for vision-centric tasks such as image generation, image captioning, and visual question answering, where language plays a secondary role. However, we are confident that the limitations imposed by visual priors on language-heavy tasks can be effectively addressed through scaling. As shown in our responses to Q1 and Q2, increasing the amount of training data already yields promising improvements, suggesting strong potential for further gains through model scaling.
>
> Moreover, recent works such as DeepSeek-OCR [1] and Glyph [2] illustrate an alternative perspective on handling language-heavy tasks: representing language through images. Taken together, these insights indicate that with larger-scale visual-prior models and increased data and compute, we are confident that larger visual prior models can continue to close the gap on language-heavy tasks.
>
> * [1] DeepSeek-OCR: Contexts Optical Compression: https://arxiv.org/abs/2510.18234
> * [2] Glyph: Scaling Context Windows via Visual-Text Compression, https://arxiv.org/abs/2510.17800

---

### Official Review · Reviewer_9HJx · 2025-11-02

**Soundness:** 3
**Presentation:** 3
**Contribution:** 3
**Rating:** 6
**Confidence:** 3

**Summary:**

This paper presents Muddit, a unified discrete diffusion model that bridges text and image generation within a single architecture. By leveraging strong visual priors from a pretrained text-to-image backbone (Meissonic) and employing fully discrete diffusion for both modalities, Muddit achieves competitive performance on text-to-image generation, image captioning, and visual question answering tasks. The work demonstrates that discrete diffusion, when equipped with appropriate pretrained knowledge, offers a scalable and efficient alternative to autoregressive approaches for unified multimodal generation.

**Strengths:**

1. The method design is novel. It achieves multimodal unification through fully discrete diffusion for both text and image, unlike hybrid approaches that combine autoregressive and diffusion components with separate architectures.

2. The method enables parallel token generation, achieveing speedup over AR baselines. This is good for evaluation and real-time applications.

3. The paper is well-written and easy to follow.

**Weaknesses:**

1. The key designs of this work (Initializing from a visual prior rather than LLM, using CLIP as text encoder rather than LLM) fundamentally constrain the model's text comprehension capabilities. This is particularly evident in handling long-form, complex text. These limitations restrict the practical applicability of the method, especially in scenarios requiring sophisticated natural language understanding or generation beyond simple captions and short answers.

2. Text generation is constrained to 77 tokens, limiting applicability for long-form generation tasks.

3. The experiment results are not very strong: there is a gap with AR baselines in several tasks (e.g. VQAv2), and with continuous diffusion models in generation task.

**Questions:**

Do you study how the approach scale with model size, training data, and computation resource? I am curious about whether the performance gap can be closed through scaling.

---

> ### Author Response · Authors · 2025-11-26
> **Response to Reviewer 9HJx**
>
> > W1. The key designs of this work (Initializing from a visual prior rather than LLM, using CLIP as text encoder rather than LLM) fundamentally constrain the model's text comprehension capabilities. This is particularly evident in handling long-form, complex text. These limitations restrict the practical applicability of the method, especially in scenarios requiring sophisticated natural language understanding or generation beyond simple captions and short answers.
>
> Thanks for your insightful feedback. We respectively want to clarify that visual priors inherently necessitate strong text-following capabilities to handle long-form or complex textual inputs, demonstrating a robust correlation between the image and text modalities. From a unified model perspective, we prioritize this cross-modal alignment over isolated single-modality performance. Furthermore, during the rebuttal, we scaled up Muddit by incorporating 6M additional pretraining and instruction-tuning data and computational resources. The clear gains suggest that the understanding gap compared to LLMs can be effectively mitigated through data and computation scaling.
>
> | Model | Params | Base model | Architecture | Data scale | VQAv2 | MME | MMMU |
> | :--- | :--- | :--- | :--- | :--- | :--- | :--- | :--- |
> | Show-o (512 x 512) | 1.3B | Phi-1.5 | AR + Discrete Diff | 35M | 69.4 | 1097.2 | 27.4 |
> | D-DiT (512 x 512) | 2B | SD3-medium | Discrete Diff + Diff | 40M | 60.1 | 1124.7 | - |
> | **Muddit (1024 x 1024)** | **1B** | **Meissonic** | **Discrete Diff** | **16M** | **70.2** | **1139.2** | **28.7** |
>
> > W2. Text generation is constrained to 77 tokens, limiting applicability for long-form generation tasks.
>
> We acknowledge that Muddit's current base model constrains the length of text generation. We identify two primary solutions to address this. The first is to replace the text encoder with a long-context model, such as Gemma-2; however, this approach requires substantial data and computational resources for retraining. A second, more immediate solution is to employ long-form generation techniques [1, 2] designed for language discrete diffusion models. Since Muddit is a fully discrete diffusion model, these methods are inherently compatible and can effectively extend its text generation capabilities.
>
> * [1] M Arriola et al. Block Diffusion: Interpolating Between Autoregressive and Diffusion Language Models, ICLR 2025 Oral, https://arxiv.org/abs/2503.09573
> * [2] S Nie et al. Large Language Diffusion Models, https://arxiv.org/abs/2502.09992
>
> > W3. The experiment results are not very strong: there is a gap with AR baselines in several tasks (e.g. VQAv2), and with continuous diffusion models in the generation task.
>
> In image understanding tasks, by scaling data and computational resources, we achieved performance superior to comparable AR-based models across multiple benchmarks while utilizing less data. This validates both the feasibility and the data efficiency of our visual prior approach. Regarding image generation, Muddit demonstrates a substantial improvement over the base model, rising from 0.54 to 0.62 (15%). In contrast, continuous diffusion methods like D-DiT show only marginal gains, increasing from 0.62 to 0.65 (4.8%). Furthermore, by employing test-time scaling, Muddit achieves a GenEval score of 0.67, surpassing the 7B parameter MMaDA model.
>
> | Model | Params | Base model | Data scale | Geneval w TTS |
> | :--- | :--- | :--- | :--- | :--- |
> | MMaDA ($512 \times 512$) | 8B | LLaDA | Unknown | 0.66 |
> | **Muddit ($1024 \times 1024$)** | **1B** | **Meissonic** | **16M** | **0.67** |
>
> > Q1. Do you study how the approach scale with model size, training data, and computation resource? I am curious about whether the performance gap can be closed through scaling.
>
> As mentioned in Q1, to demonstrate the scalability of our approach, we incorporate an additional 6M samples and report the resulting performance improvements in the table. This dataset includes 2M image–caption pairs and 4M VQA pairs across diverse domains, including daily activities, animals, art, science, and reasoning. Adding the 2M image–caption pairs expands the pretraining data to 10M samples (8M + 2M), while the 4M VQA pairs increase the instruction-tuning data to 6M samples. Muddit is trained with a batch size of 512 at a resolution of 1024. Evaluation on established benchmarks shows consistent gains in both image generation and image understanding tasks, empirically validating the scalability of our model.
>
> | Model | Params | Data scale | Geneval | VQAv2 | MME | MMMU |
> | :--- | :--- | :--- | :--- | :--- | :--- | :--- |
> | Muddit ($512 \times 512$) | 1B | 10M | 0.61 | 68.2 | 1107.4 | 27.6 |
> | **Muddit ($1024 \times 1024$)** | **1B** | **16M** | **0.62** | **70.2** | **1139.2** | **28.7** |

---

### Author Response · Authors · 2025-11-26
**General Response**

# General Response

We sincerely thank all reviewers for their thoughtful and constructive feedback. We also appreciate the reviewers’ helpful suggestions for improvement. We have tried our best to revise the paper in response to all concerns.

## Strengths Highlighted by Reviewers

* **Novelty of the Framework:** Reviewers 9HJx, TaRs, and aAn6 commended the novelty of our approach, particularly the use of visual priors and a single unified framework for multimodal modeling.
* **Strong Performance of Muddit:** Reviewers G1Hq, aAn6, and TaRs recognized that Muddit achieves performance comparable to, and in some cases surpassing, LLM-based unified models, highlighting the strength of our non-LLM design.
* **Efficiency Advantages:** All reviewers acknowledged the significant efficiency benefits of Muddit compared with autoregressive baselines.

## Addressing Reviewer Concerns

### Scalability of Muddit (Reviewers 9HJx, TaRs, aAn6)
To address scalability concerns, we extended training with an additional 6M data samples. The updated results demonstrate clear, consistent improvements as we scale both data and compute, validating Muddit's scalability. We have updated our manuscript for the scaling experiments in Section 3.5, Tables 6 and 7.

### Language Understanding of Visual Prior (Reviewers 9HJx, TaRs)
We address this concern through:

**Data scaling:** With larger training data, the performance gap between visual priors and language priors on image understanding tasks is substantially reduced.

**Efficiency:** Visual priors achieve strong image understanding performance with significantly less training data; we attribute this to their strong multimodal interaction in image generation.

The experimental details and results are presented in the revised manuscript, Section 3.5, Tables 6 and 7.

### Advantages of the Discrete Diffusion Model and the Distinction of Muddit (Reviewer G1Hq)
We address this concern through:

**Performance:** Muddit shows a clear advantage over other unified models of comparable size, demonstrating the feasibility of using discrete diffusion to build a unified multimodal model.

**Data Efficiency:** Muddit surpasses other comparable unified models with significantly less training data. We attribute this to our unified modeling approach, which simplifies the training process and makes a visual prior discrete unified model a promising path.

**A New Path Toward Unified Models:** We extend discrete diffusion models to a broader set of image understanding tasks and show that a unified, visual prior, fully discrete diffusion model can be both effective and data efficient on such tasks—beyond their traditional use in text-to-image generation. This represents a core contribution of our work.

The discussion is presented in the revised manuscript, Section 3.5.

### Experimental Details (Reviewer aAn6)
We have added further experimental details in Appendix C.

---

In addition, we provide detailed responses to all reviewers' questions.
We thank the reviewers again for helping us strengthen the paper. We are always looking forward to open discussions. We will give our response as soon as possible once you raise more questions.

Sincerely,
Authors of Muddit

---

### Meta-Review · Area_Chair_D72d · 2026-01-07

**Summary:**

This paper constructs a unified discrete diffusion model that bridges text and image generation within a single architecture, by leveraging strong visual priors from a pretrained text-to-image backbone. The resulting model achieves competitive performance with existing AR or diffusion or AR/diffusion mixture models at relatively small model size. While some reviewers raised important concerns, they are reasonably addressed by the rebuttal. I'm therefore delighted to recommend acceptance.

**Reviewer Concerns:**

I feel most concerns are addressed.

**Reviewer Scores:**

Just my guess: G1Hq and TaRs might increase their scores.

---

### Decision · Program_Chairs · 2026-01-26

Accept (Poster)